# What Comes Next? Evaluating Uncertainty in Neural Text Generators Against Human Production Variability

**Mario Giulianelli**⚲* **Joris Baan**⚲*
**Wilker Aziz**⚲ **Raquel Fernández**⚲ **Barbara Plank**▲⊘☕
⚲ University of Amsterdam ⊘ ITU Copenhagen ▲ MCML Munich ☕ LMU Munich
{m.giulianelli,j.s.baan,raquel.fernandez,w.aziz}@uva.nl, b.plank@lmu.de

## Abstract

In Natural Language Generation (NLG) tasks, for any input, multiple communicative goals are plausible, and any goal can be put into words, or *produced*, in multiple ways. We characterise the extent to which human production varies lexically, syntactically, and semantically across four NLG tasks, connecting human production variability to *aleatoric* or *data uncertainty*. We then inspect the space of output strings shaped by a generation system's predicted probability distribution and decoding algorithm to probe its uncertainty. For each test input, we measure the generator's calibration to human production variability. Following this *instance-level* approach, we analyse NLG models and decoding strategies, demonstrating that probing a generator with multiple samples and, when possible, multiple references, provides the level of detail necessary to gain understanding of a model's representation of uncertainty.[1]

## 1 Introduction

Humans display great variability in language production, in particular when the context or the task are open-ended, such as in storytelling or in dialogue. Given a story prompt, for example, there are many plausible ways in which different humans (or a single writer, if asked multiple times) may tell the story (Fan et al., 2018). We refer to this phenomenon as *production variability*. Production variability in humans has two main sources. First, when situated in a context, speakers may entertain variable communicative goals (Searle, 1969; Sacks et al., 1974; Austin, 1975), and the number and variety of plausible communicative goals depends on the production task (Jokinen, 1996). Translation, for instance, defines the communicative goal almost unequivocally while a dialogue context might allow for a wide variety of communicative goals (expressed, *e.g.*, as a request, an assertion,

---

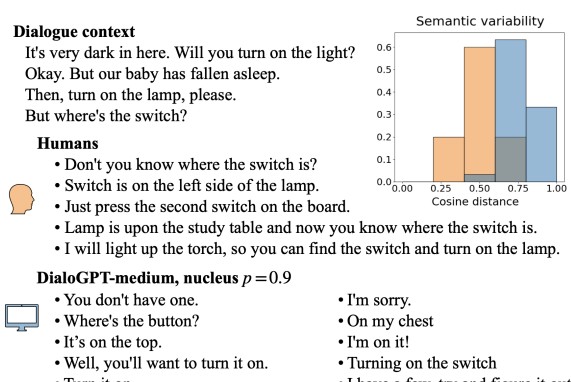

**Dialogue context**
It's very dark in here. Will you turn on the light?
Okay. But our baby has fallen asleep.
Then, turn on the lamp, please.
But where's the switch?

**Humans**
- Don't you know where the switch is?
- Switch is on the left side of the lamp.
- Just press the second switch on the board.
- Lamp is upon the study table and now you know where the switch is.
- I will light up the torch, so you can find the switch and turn on the lamp.

**DialoGPT-medium, nucleus $p = 0.9$**
- You don't have one.
- Where's the button?
- It's on the top.
- Well, you'll want to turn it on.
- Turn it on.
- I'm sorry.
- On my chest
- I'm on it!
- Turning on the switch
- I have a few, try and figure it out.

Figure 1: Production variability observed in 5 human responses vs 10 responses generated by DialoGPT. The graph presents the distribution of pairwise cosine distances: generated responses exhibit higher semantic variability than human responses. The generator's semantic uncertainty is too high in this dialogue context.

---

or a yes-no question). The second source of variability is the fact that even when context and communicative goal are fixed, speakers' linguistic realisations of the communicative goal may vary (Levelt, 1993). Both sources of variability apply to individuals as well as to populations: if an expert is asked to simplify a complicated sentence multiple times, they may perform different rewriting transformations (*e.g.*, paraphrasing, reordering, or sentence splitting) and produce different texts (Alva-Manchego et al., 2021); the same is true if multiple experts are asked to perform a task (Xu et al., 2015). If we are to regard a Natural Language Generation (NLG) system (or *text generator*) as a good model of human production, it should capture the variability observed in humans.

Text generators combine two mechanisms: (i) an underlying statistical model—typically, an autoregressive factorisation of the probability of sequences, with conditional token probabilities predicted by a neural network; and (ii) an iterative decoding algorithm that chains samples from next token distributions into a complete production. To-

---

*Equal contribution.

[1]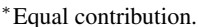https://github.com/dmg-illc/nlg-uncertainty-probes

gether these two mechanisms specify a probability distribution over sequences of tokens, which can be regarded as a representation of the model's uncertainty about productions for a given generation context (see Baan et al. (2023) for a detailed discussion). In this work, we assess whether this representation of uncertainty is in compliance with production variability exhibited by a population of humans—which in turn, we argue, can be regarded as an expression of *aleatoric* uncertainty, *i.e.*, irreducible uncertainty due to the stochastic nature of the data generating process (Der Kiureghian and Ditlevsen, 2009; Hüllermeier and Waegeman, 2021). In other words, we compare the distribution over productions of a text generator against the distribution over the productions of a population of human speakers, given the same context (Figure 1).

Quantifying the closeness in distribution between a text generator and a human population is difficult: we only have an iterative view into the generator's distribution; the 'human distribution' is an implicit or even hypothetical object; and in both cases, the sample space is large or even unbounded. We can, however, compare these two objects via the samples they produce and assess their statistical distance—which is what we propose here. For each individual generation context, we compare scalar properties of generations (through repeated model sampling) and human productions (using multi-reference NLG datasets). In particular, we *probe* for lexical, syntactic, and semantic distance between productions, thus allowing for a quantitative and interpretable assessment of uncertainty.

We find that the uncertainty of neural text generators is higher than justified by human production variability in open-ended tasks, like story generation and open-domain dialogue; and that it is lower on more constrained tasks, like machine translation and text simplification. Popular decoding algorithms, which bias away from the distribution of the generator's underlying statistical model (*e.g.*, top-$k$, top-$p$, or locally typical, rather than ancestral sampling), have a limited impact on the generator's ability to faithfully represent human variability. We complement our quantitative assessments with a detailed analysis of individual generation contexts, which sheds light on whether a generator has robustly learned to reproduce degrees and aspects of human variability plausible for the communicative task.

Beyond the experimental results obtained on our selection of models and tasks, our work has important implications for NLG evaluation and data collection. Multiple samples and, when possible, multiple references, should be used to assess the statistical fit of text generators. Our approach, complementary to other types of automatic evaluation, makes model assessments particularly insightful and trustworthy because it does not judge a model only by a single output but also, intuitively, by *what it could have generated*—and it does so for each individual input in the test set. We therefore hope our framework will be used by the community as an evaluation criterion for NLG systems, especially to assess them in more open-ended tasks.

## 2  Related Work

Automatic approaches to the evaluation of NLG systems are of high practical importance: they allow for model selection at scale and power quality-aware decoding algorithms (Borgeaud and Emerson, 2020; Eikema and Aziz, 2020; Fernandes et al., 2022; Suzgun et al., 2022). In spite of their known limitations (Gehrmann et al., 2022), they are a necessary complement to human evaluation (Belz and Reiter, 2006; van der Lee et al., 2021).

**Reference-based evaluation.** The most common way of automatically evaluating text generators is via metrics that estimate the similarity between candidate generations and references, such as BLEU (Papineni et al., 2002), ROUGE (Lin, 2004), COMET (Rei et al., 2020), BLEURT (Sellam et al., 2020), and BertScore (Zhang et al., 2020a). Reference-based metrics are less suited for open-ended tasks such as story generation and dialogue, where a single reference (or even a handful) cannot be representative of the large space of plausible communicative goals and realisations.

**Reference-free evaluation.** A popular, reference-free alternative is to train evaluation models that discriminate human from model output (*e.g.*, Bruni and Fernández, 2017; Gehrmann et al., 2019; Hashimoto et al., 2019), score the appropriateness of input-output pairs (*e.g.*, Sinha et al., 2020; Fomicheva et al., 2020), or model human judgements directly (*e.g.*, Lowe et al., 2017; De Mattei et al., 2021; Rei et al., 2021). Neural language models themselves have been proposed as evaluators (*e.g.*, Yuan et al., 2021; Deng et al., 2021) and used to assess generations along interpretable evaluation dimensions (Zhong

et al., 2022), yet they have been criticised for being biased (toward models similar to the evaluator) and thus limited in their ability to evaluate generated text (Deutsch et al., 2022).

**Statistical evaluation.** Statistical evaluation compares model generations to human productions *in distribution* through real-valued statistics (*e.g.*, Zipf's coefficient, type-token ratio, length) as opposed to strings themselves. These statistics are typically compared marginally, at the corpus level (Eikema and Aziz, 2020; Meister and Cotterell, 2021; Pillutla et al., 2021; Pimentel et al., 2022), supporting general claims about model performance in relation to humans. More recently, Barkhof and Aziz (2022) and Deng et al. (2022) compared statistics at the instance level, supporting claims about models' performance in relation to humans for individual inputs. In this work, we craft statistics that evaluate generators' uncertainty at the *instance level* against the variability over sequences observed in multi-reference NLG datasets. Although evaluating uncertainty is gaining traction in NLP (*e.g.*, Desai and Durrett, 2020; Glushkova et al., 2021; Baan et al., 2022), there is relatively little work on sequence-level uncertainty (Ott et al., 2018; Malinin and Gales, 2020; Aina and Linzen, 2021; Kuhn et al., 2022).

**Diversity in NLG.** Our analysis is related to NLG studies on output diversity. Some have evaluated diversity induced by different models and NLG decoding strategies—yet do not use human levels of variability as a target (Wiher et al., 2022)—or have used human judgements to evaluate diversity metrics themselves (Tevet and Berant, 2021; Stasaski and Hearst, 2022). Others have developed diversity-enhancing objective functions (Li et al., 2016) and decoding algorithms (Vijayakumar et al., 2018; Shu et al., 2019; Weir et al., 2020; Meister et al., 2021). In our study, where the aim is to evaluate the uncertainty of NLG systems, we focus on unbiased sampling and the most widely used decoding algorithms.

## 3 Probing Language Processes for Production Variability

We interpret language production, by humans or NLG systems, as captured by a probability distribution over natural language strings (*productions*), a random variable $Y$, given a linguistic context $X = x$. The context $x$ can be a source sentence

in translation, a story prompt in story generation, or more generally the *input* to a language process. In turn, a production is a piece of text $y$ such as a single translation, a story, or more generally the *output* of a language process.[2]

### 3.1 Production Variability

For any language process, *production variability* is fully characterised by a conditional probability distribution $p_{Y|X=x}$ representing uncertainty about the output $Y$ given input $X = x$. Intuitively, the uniform distribution maximises production variability and the Dirac delta (one-hot) distribution minimises it. Analysing this distribution is difficult. Notably, for human language processes, we do not have an explicit representation of $p_{Y|X=x}$. This prevents a direct comparison through measures of statistical divergence, or summaries like entropy. Through data collection we can, however, draw conditional samples from the human language process (*i.e.*, gather references given a context). On the other hand, for NLG models, we do have an algorithmic representation of $p_{Y|X=x}$, which is usually sufficient to enable sampling, but the unbounded sample space and lack of conditional independence assumptions make statistical divergence and summaries like entropy intractable.[3]

Instead, we propose to analyse language processes through their samples. This in turn introduces other difficulties, as text is a high-dimensional, structured, non-numerical data type. For tractable analysis, we exploit a set of real-valued and interpretable statistics, or *production probes*, to re-express a language process distribution in terms of how, given an input, its outputs relate to outputs of another language process. When both processes are independent humans performing a task, we obtain a sense of how plausible human productions relate (or *vary* with respect) to other plausible human productions, along a linguistically interpretable dimension. When we swap one or both processes for an NLG model, we obtain tools to analyse how model generations relate to plausible human productions, thus assessing a model's representation of uncertainty against the variability observed in humans.

---

[2]**Notation.** Random variables are denoted by uppercase letters (*e.g.*, $Y$), outcomes are lowercased (*e.g.*, $y$), and $p_{Y|X=x}$ denotes the probability distribution of $Y$ given $X = x$.

[3]In Appendix A, we discuss issues with entropy in more detail—how it is both difficult to estimate and interpret for text generators.

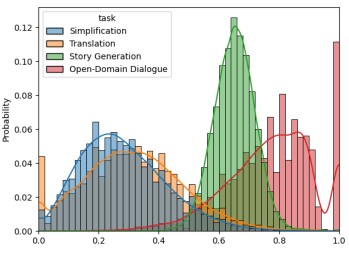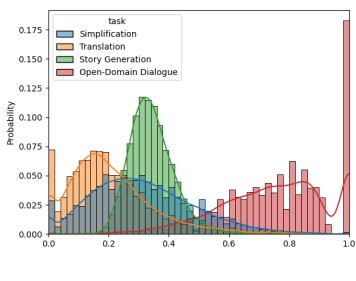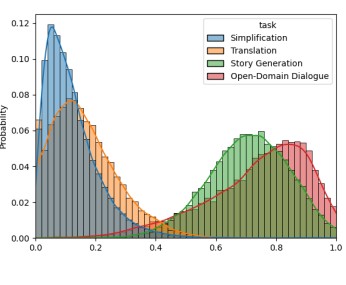

| (a) Lexical variability | (b) Syntactic variability | (c) Semantic variability |

Figure 2: Human production variability across four NLG tasks. The values on the horizontal axis are single samples of lexical (unigram), syntactic (POS bigram), or semantic (cosine) distance between two randomly sampled productions for each input (see Section 3). Re-sampling productions results in nearly identical marginal distributions. Probability mass on the right side signals high distance and thus high variability, and vice versa.

Specifically, given a context $x$, two language processes with distributions $p_{\hat{Y}|X=x}$ and $p_{Y|X=x}$, and a choice of distance metric $k(\cdot, \cdot) \in \mathbb{R}$, *our probe for production variability* is a real random variable $k(\hat{Y}, Y)$. This random variable captures the joint distribution of distance between any two outputs drawn conditionally from the two processes. The distribution of the probe $k(\hat{Y}, Y)$ is also intractable, but we can estimate it via simulation by drawing productions from the processes and assessing the distance metric on sampled pairs, as illustrated in Figure 1.

Consider analysing the human process (§ 5) through $k(Y, Y)$: when multiple realisations of the output are dissimilar (*e.g.*, given the input '*How is your day?*' and outputs '*Fantastic, thank you!*' and '*I asked you first*') production variability is high along the dimension captured by $k$.

### 3.2 Production Probes

We instantiate our production probes with three distance functions. They return values from 0 to 1. We hope that future work will experiment with alternative probes that may capture other linguistic or extra-linguistic levels of analysis.

**Lexical:** The fraction of distinct $n$-grams in two strings, with $n \in [1, 2, 3]$ (*i.e.*, number of non-matching $n$-gram occurrences divided by the total number of $n$-grams in both strings).

**Syntactic:** Analogous to lexical distance but on part-of-speech tag $n$-grams.[4]

**Semantic:** The cosine distance between the sentence embeddings of two strings (Reimers and Gurevych, 2019).[5]

---

[4] From spaCy, en_core_web_md (Honnibal et al., 2020).
[5] `sentence-transformers/all-distilroberta-v1`

## 4 Experimental Setup

### 4.1 Data and Models

We experiment with four NLG datasets that contain 5+ human references per input instance and for which we expect humans to display different degrees of production variability. For each tasks, we select models that are publicly available, are reasonably sized, have been used previously on the task, and are conventionally accepted as suitable for it.[6] All datasets are in English; for translation, the target language is German. Table 1 (Appendix C) shows relevant statistics. The reference collection procedure varies across datasets and we discuss how this may impact our analysis in the Limitations section.

**Machine translation.** We use 500 sentences from the WMT-14 En-De test set (*newstest2014*; Bojar et al., 2014), which have been annotated by Ott et al. (2018) with 10 additional reference translations produced by as many human translators. As a generator, we use Helsinki-NLP's Transformer-Align model trained on Opus-MT (Tiedemann and Thottingal, 2020).

**Text simplification.** We use the 2,000 instances of the *ASSET* validation set (Alva-Manchego et al., 2020). For each source sentence, originally from the TurkCorpus (Xu et al., 2016), ASSET includes 10 additional simplifications by as many crowdsource annotators. On this dataset, we test Flan-T5-large (Chung et al., 2022), an instruction-finetuned version of the T5 language model (Raffel et al., 2020), which we further finetune on the ASSET training set.

---

[6] For text simplification, we did not find any available trained model, so we used a versatile model like FlanT5.

**Storytelling (story generation).** We use the 759 instances from the *WritingPrompts* test set (Fan et al., 2018) for which at least 5 human references are available. Prompts and stories are originally scraped from r/WritingPrompts, a Reddit forum of stories written by online users in response to story prompts designed by other users. The number of stories available per prompt (9.56 ± 7.67) varies from 5 to 92. We use GPT2-large (Radford et al., 2018) finetuned on the WritingPrompts training set.

**Open-domain dialogue.** We use the development set of *DailyDialog++* (Sai et al., 2020), which contains 5 additional references for 1,028 conversations from the DailyDialog corpus (Li et al., 2017). The dialogues are short (less than 8 turns) and cover a broad list of topics; for each dialogue, 2-3 annotators were asked to generate 1-3 alternative responses.[7] For this task, we use the pretrained DialoGPT-medium (Zhang et al., 2020b).

### 4.2 Decoding algorithms

We experiment with five decoding algorithms: unbiased (ancestral or forward) sampling (Bishop, 2006; Koller and Friedman, 2009), temperature scaling, top-$k$ sampling (Fan et al., 2018), nucleus sampling (Holtzman et al., 2019), and locally typical sampling (Meister et al., 2023). For all decoding algorithms, we set the maximum sequence length to 100 (cf. Table 1, Appendix C).

## 5 Human Production Variability Across NLG Tasks

Consider $p_{Y|X=x}$ the distribution that describes the human language process, and define the following special case for human production variability:

$$H_k(x) \coloneqq k(Y, Y) . \tag{1}$$

Estimating this probe by drawing pairs of human productions provides an interpretable view on plausible variability—*i.e.*, aleatoric uncertainty—along the dimension captured by $k$. Figure 2 shows $H_k(x)$ marginalised over inputs for the four NLG tasks. We use unigram distance for the lexical probe, POS bigram distance for the syntactic probe, and cosine distance for the semantic probe. High distance indicates high variability, and vice versa.

---

[7] The *DailyDialog++* annotators are also instructed to avoid short generic responses such as '*Sure*' and to write, instead, meaningful responses with at least 8-10 words.

**Translation and text simplification.** Humans show low production variability in these two tasks. While translations of a given source sentence are more lexically and semantically varied, simplifications exhibit a higher degree of syntactic variability, probably as a result of the instructions used during data collection (writers were asked to use varying rewriting transformations). Overall, low levels of variability are to be expected as, in both tasks, content preservation is part of the communicative goal.

**Story generation.** Variability in story generation is strongly dependent on the probe. It is low at the syntactic level—close to translation and simplification—while lexical and semantic probes place this task closer to open-domain dialogue. Stories generated from a given prompt may vary a lot in content, but basic syntactic structures and lexical material are shared. Although this task can be a priori perceived at least as 'open-ended' as dialogue, lower levels of variability may result from contextual factors specific to the *WritingPrompts* dataset that we are not explicitly modelling, such as writers reading stories contributed by other users.

**Open-domain dialogue.** We observe the highest production variability in this task across all probes. Many output pairs are lexically and syntactically completely dissimilar, as indicated by the rightmost bin in Figures 2a and 2b. Lexical variability is even more extreme when looking at bigrams and trigrams (Figure 7 in Appendix D) suggesting that while responses rarely share words or phrases, they still sometimes convey similar meaning (Figure 2c). Overall, the fact that dialogue appears to be the most open-ended task can be explained by the wide variety of communicative goals that can plausibly follow from a dialogue context and, in part, by the fact that individual annotators produced multiple responses for the *DailyDialog++* dataset and thus were able to monitor the diversity of their outputs.

## 6 Do Neural Text Generators Reproduce Human Production Variability?

Consider, now, a second language process: a text generator with distribution $p_{\hat{Y}|X=x}$. We study this generator's uncertainty about outputs given an input $x$ under *two lenses*. In § 6.1, we study how outputs vary with respect to one another, which is analogous to human production variability $H_k(x)$. We refer to this as the generator's *self-variability*:

$$M_k(x) \coloneqq k(\hat{Y}, \hat{Y}) . \tag{2}$$

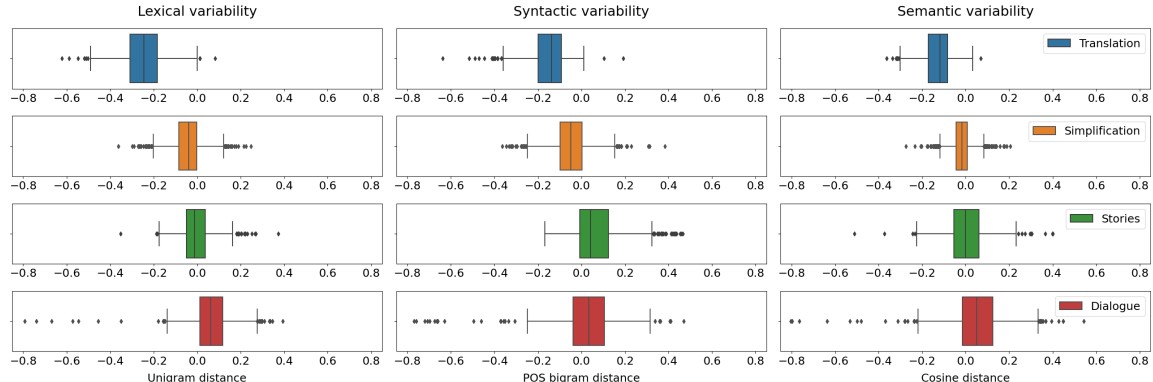

Figure 3: Distribution of $\mu_{M_k(x)} - \mu_{H_k(x)}$ over instances. Values greater than zero indicate the model overestimates the variability of the task (higher mean pairwise distance); values below zero indicate variability underestimation.

In § 6.2, instead, we study how model generations vary with respect to a language process *known to be plausible*: a human language process $p_{Y|X=x}$. We refer to this as *cross-variability*:

$$C_k(x) \coloneqq k(\hat{Y}, Y) \, . \tag{3}$$

Our expectation is that generators with a good representation of aleatoric uncertainty reproduce human production variability along both axes. As we employ a distance metric, it may look like we should regard a model as a good approximation to the human process whenever $C_k(x)$ concentrates about small positive values. To some extent this is the interpretation exploited by most automatic evaluation metrics (single- or multi-reference). In this work, we refrain from taking any one human production as a 'reference' to be closely 'matched'; rather, we take statistical properties of human productions as illustrative of plausible variability and thus targets to be reproduced. We quantify deviation from plausible human variability by estimating a notion of statistical divergence.

## 6.1 The Underlying Statistical Model

In this section, we criticise the underlying statistical model (as a result of parameter estimation via MLE) using unbiased sampling. As models observe variability only marginally (multiple references are rarely used during training), it is interesting to study if their self-variability is calibrated to human variability: given individual input instances, do distances between unbiased model samples distribute similarly to distances between human productions? To distinguish over-estimation from under-estimation of variability, we report a signed notion of divergence, $\mu_{M_k(x)} - \mu_{H_k(x)}$. When $M_k(x)$ and $H_k(x)$ distribute similarly, their mean

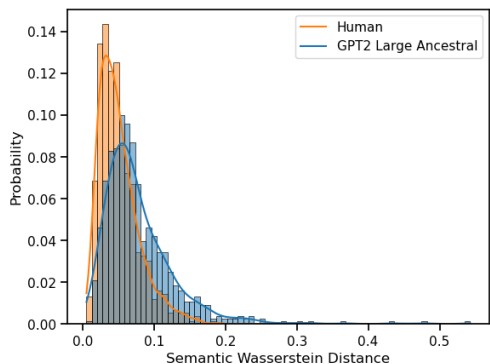

Figure 4: Distribution over Wasserstein distances for GPT-2 in blue: $\mathrm{D}_{W_1}(C_k(x), H_k(x))$. Distribution for a human control group in orange: $\mathrm{D}_{W_1}(\hat{H}_k(x), H_k(x))$. Semantic probe: $k$ is cosine distance.

difference is low for a given $x$. Positive differences imply that models overestimate variability, *i.e.*, model samples vary more with respect to one another than human samples do. Negative differences indicate that models underestimate variability.

Figure 3 shows how mean differences distribute across each task-specific test set for the models in Section 4. We use up to 10 human productions (5 for dialogue) and 10 generations. The first two rows show that $\mu_{M_k(x)} - \mu_{H_k(x)}$ distributes far below 0 for translation (OpusMT) and somewhat below 0 for simplification (Flan-T5), indicating that the two models substantially underestimate variability.[8] The opposite is true for dialogue and story generation: both GPT-2 and DialoGPT moderately overestimate the open-endedness of these tasks. We also inspect cross-variability $\mu_{C_k(x)} - \mu_{H_k(x)}$, finding similar patterns, with slightly better over-

---

[8] OpusMT uses label smoothing, which is known to harm the distribution of unbiased samples along dimensions such as $n$-gram and skip-bigram counts (Eikema and Aziz, 2020).

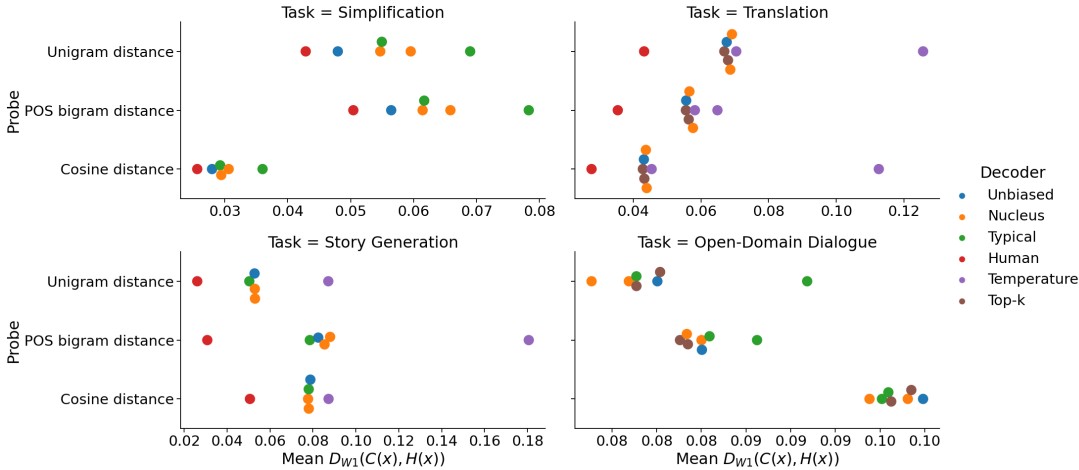

Figure 5: Mean Wasserstein distances $D_{W_1}(C(x), H(x))$ for (tasks, probe, decoder) tuples. Base models for each task are described in Section 4. Colour refers to decoding algorithm with various parameter settings (fully reported in Table 4, Appendix F). Human control group in red.[11] Clusters suggest that decoders often have similar effect. Unbiased sampling is competitive.

all cross-variability calibration for translation and simplification (Figure 8, Appendix D).

## 6.2 The Effect of Decoding Algorithms

We now study text generators obtained by varying the sampling procedure.[9] We analyse their representation of uncertainty by assessing the divergence between the distribution of generator-human cross-variability $C(x)$ and human variability $H(x)$. While $\mu_{C_k(x)} - \mu_{H_k(x)}$ can inform us about the direction of miscalibration, we observe only a handful of cases where different decoding strategies yield both under- and over-estimation for the same model (see Figures 10 and 11 in Appendix D). Instead, as we sometimes observe distributions with multiple modes—causing their difference in means to paint an incomplete picture—we additionally report a measure of divergence that is more robust to such multi-modal distributions: the Wasserstein 1-Distance $D_{W_1}(\cdot, H_k(x))$.[10] Results for self-variability $M(x)$ and mean distance can be found in Appendix D, Figures 9 to 11.

**Human control group.** The blue curve in Figure 4 shows how $D_{W_1}(C_k(x), H_k(x))$ distributes over inputs for unbiased samples from GPT-2 on story generation. To contextualise this observation

we report a human control group (the orange curve): this is $D_{W_1}$ measured between two human populations (*i.e.*, we make two disjoint samples from the available human productions for each prompt, use those to estimate $H_k(x)$ and an analogous $\hat{H}_k(x)$, and compute $D_{W_1}(\hat{H}_k(x), H_k(x))$). We can now appreciate what is a plausible Wasserstein distance curve between two human-based processes, and with that, we can better discern that this particular system gives good but not perfect representation to human levels of production variability (note the overlap between the two distributions). Upon visual inspection of divergence distributions (like Figure 4) for different sampling strategies, we find similar shapes. We exploit this finding and summarise each divergence distribution using its mean. This is shown in Figure 5, which presents results for many decoding settings, tasks and probes. The leftmost red dots indicate the human control group.[11] We observe that two human groups agree more on the meaning of translations and simplifications than on their form, while for story generation the two groups agree more on surface form and basic structures and less on the semantic content of the stories.

**Results.** Overall, Figure 5 shows that most decoding settings are close to unbiased sampling, which in turn is in the same ballpark (mean Wasserstein distance always lower than 0.1) as the human control. This indicates that text generators

---

[9]This leads to a probability distribution whose pmf is hard if at all possible to characterise, meaning we cannot easily assess the probability of an outcome under the new distribution. But we have an explicit sampler for this new distribution, which is all our analysis tools require.

[10]Indeed, we find several cases where $D_{W_1}$ signals stronger miscalibration compared to $D_\mu$. For an additional discussion about $D_{W_1}$, see Appendix B.

[11]No control condition is shown for open-domain dialogue as the five references contained in *DailyDialog++* are too few to create a control group.

capture the space of plausible human productions well when coupled with most decoding algorithms, though not as well as another human language process. Decoding settings form many clusters, and for all tasks except open-domain dialogue, unbiased samples best match human variability. This suggests that, within the limits of decoding configurations typically considered as appropriate, different token-level decoding strategies often have a similar effect on a generator's ability to reproduce human production variability along our three probes. Altogether, these findings inform us about an often neglected aspect of decoding algorithms, namely their effect on the model's representation of uncertainty (rather than their ability to select individual high-quality generations).

## 7 Qualitative Instance-Level Analysis

We now qualitatively analyse individual inputs for which a generator's uncertainty is miscalibrated to human variability—as detected by $D_{W_1}$. For each task, we use up to 10 human productions (5 for dialogue) and 10 generations. Figures accompanying the examples in this section are in Appendix E. While it is not a replacement for more standard NLG evaluation procedures, we argue that this level of analysis is complementary and crucial to gain deeper understanding of a generator's representation of uncertainty.

**Variability underestimation in translation and simplification.** We have seen that in translation and simplification, generators' self-variability is lower than human variability (§ 6.1). We now zoom in on examples from these two tasks, inspecting instances that show inadequate model fit on all linguistic levels (*i.e.*, $D_{W_1}(M_k(x), H_k(x))$ is high for all $k$). The most severe cases of miscalibration for OpusMT are all instances of variability underestimation.[12] For most of these, generations are virtually or completely identical, while a few present slightly higher but still substantially lower variability than human productions. For example, ten humans translated the phrase '*reacted cautiously*' in the English source sentence '*Several companies have thus far reacted cautiously when it comes to hiring*' in six different ways ('*vorsichtig reagiert*', '*zurückhaltend reagiert*', '*mit Vorsichtsmaßnahmen*

reagiert*', '*reagierten mit Zurückhaltung*', '*mit Vorsicht reagiert*', '*reagierten verhalten*') while all ten generated samples contain the German phrase '*vorsichtig reagiert*', signalling that the generator's lexical rephrasing abilities do not generalise to this input instance. For text simplification, we focus on instances where Flan-T5's uncertainty is not calibrated to human syntactic variability.[13] We observe that simplifications sampled from the generator are always syntactically more similar to each other than humans', indicating that the generator struggles to capture an important aspect of text simplification: that many semantically equivalent rewritings are possible if a text's syntactic structure is altered.

**Variability overestimation in dialogue.** According to our estimates of human variability (§ 5), dialogue is the most open-ended task on all linguistic levels. We have hypothesised that this is due to the large variety of communicative act types plausible given any dialogue context. We have also seen that DialoGPT generally overestimates production variability (§ 6.1)—Figure 1 is one such example. Now we further inspect instances where cross-variability is miscalibrated with respect to human outputs.[14] We find that the generator's bad fit can be due to very short and generic responses (*e.g.*, '*Well...*', '*haha*', '*Ahem*', '*Well done!*'), but is more often due to the presence of fluent yet very diverse and often inadequate samples. For such instances, not only is the generator's cross-variability miscalibrated—self-variability, too, is overestimated on all linguistic levels. In particular, the generator's poor calibration to lexical and syntactic variability is related to its inability to choose the correct dialogue acts (or favouring an excessive variety of dialogue acts). In an example instance where the last dialogue turn goes '*I've got a business call that I really need to take*', humans all reply with short affirmative responses ('*Okay! Please.*', '*Well! Go on.*', '*Sure, why not!*', '*Sure! Go ahead.*', '*Yes! Sure.*') while the model's responses are mostly lengthy statements, sometimes not particularly coherent ones (*e.g.*, '*You don't need a business call. You need a friend*').

**Variability in lack of situational grounding.** We have observed that human-written stories in the *WritingPrompts* dataset show lower variability than human dialogue responses, and hypothesised

---

[12]We select instances with $D_{W_1} > 0.3$ for unigram distance and $D_{W_1} > 0.2$ for POS bigram and semantic distance. These thresholds are chosen based on distribution plots of instance-level distances (see, *e.g.*, Figure 2b).

[13]$D_{W_1}(M_k(x), H_k(x)) > 0.2$; $k$ is POS bigram distance.
[14]$D_{W_1}(C_k(x), H_k(x)) > 0.2$ for all $k$ in {unigram distance, POS bigram distance, cosine distance}.

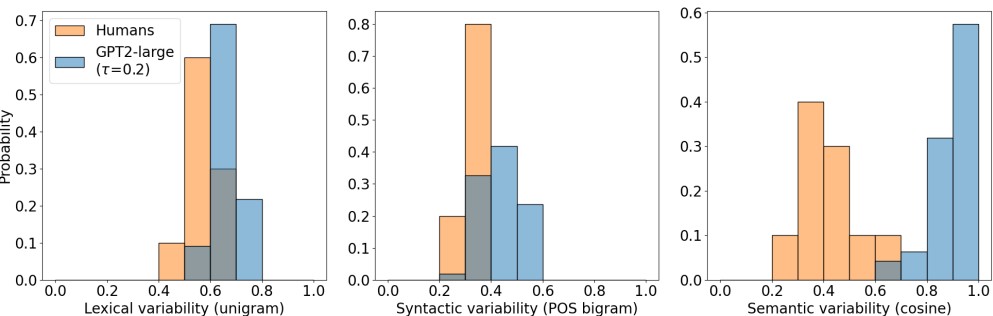

Figure 6: Example of poor cross-variability calibration for GPT-2 with typical sampling on story generation.

that this may be in part due to contextual pressures that constrain variability (§ 5). We now analyse instances flagged by our probe as cases of badly calibrated semantic cross-variability for GPT-2.[15] For one of these, the prompt refers to a portion of the situational context the model does not have access to ('*all top level comments in this prompt take place in the same world, so make them all fit together*'). Because they are conditioned on and reuse that context, human stories are quite similar to each other; generations, instead, show much higher pairwise distance both when sampled jointly with the human productions (see Figure 6) and with themselves. The lack of relevant situational grounding makes the model more uncertain than it should be for this instance.

## 8 Conclusion

Variability is an intrinsic property of human language production. Text generators, if they are to be considered as good statistical models of human written production, should exhibit plausible levels of variability. However, in NLG, the widespread practice is (i) collecting only one 'reference' production for each input and (ii) evaluating only a single generation. To appreciate the impact of this incongruity empirically, we analyse multiple-reference datasets for four NLG tasks, and show that each task has its own plausible levels of lexical, syntactic, and semantic variability. We connect production variability to aleatoric uncertainty, the irreducible uncertainty of the language production process, and evaluate neural text generators in terms of whether their representation of uncertainty is calibrated to the levels of variability observed

in humans. We find that NLG models overestimate production variability in open-ended tasks and underestimate it in more constrained tasks, and that most popular decoding algorithms all have a similar, limited effect on the generators' ability to reproduce human variability.

We advocate for more widespread usage of instance-level probing of NLG systems as a way to evaluate their statistical fit, not just along the dimensions we cover in this study but with respect to any other quality of interest. This approach contrasts with corpus-level analyses of NLG systems (*e.g.*, Pillutla et al., 2021; Meister and Cotterell, 2021; Pimentel et al., 2022) and thanks to its greater interpretability, it builds trust in the ability of generators to reproduce human-like statistics when situated in specific linguistic contexts rather than 'globally', over a possibly heterogeneous corpus. In the future, we plan to devise new ways of improving the calibration of models' uncertainty (Zhao et al., 2022; Zhang et al., 2022), *e.g.*, steering generators with sequence-level decoding algorithms (Eikema and Aziz, 2022), and to investigate the relation between uncertainty and perceived generation quality (*e.g.*, Kuhn et al., 2022): while we use human levels of variability as a target, desirable levels of variability may deviate from human statistics for specific applications.

Future work should also study production variability as a function of a more complex notion of discourse context (Giulianelli and Fernández, 2021; Giulianelli et al., 2023) and attempt to disentangle uncertainty over communicative goals and realisations (Stasaski and Hearst, 2023). This is an important avenue not only toward more practically useful generators but also toward reliable computational models of language production.

---

[15] $D_{W_1}(C_k(x), H_k(x)) > 0.3$; $k$ is cosine distance.

## Limitations

Our analysis relies on multiple-reference datasets, which are scarce for NLG tasks. Even though, for single-reference datasets, we cannot perform a similar instance-level analysis, this fact does not entail that the observations we make do not apply to such datasets—we might simply not have the data to expose them.

**Impact of data collection.** The way in which multiple references are gathered may impact the variability in productions. For example, asking a single annotator to produce several distinct references might artificially increase the diversity of responses. Conversely, asking several independent annotators might decrease diversity for they may resort to similar responses that quckly come to mind (or, in fact, the opposite if they interpret the linguistic context differently). To summarise, there are two levels of uncertainty in human production data: one is on the individual level, the other is on the population level. In this work, we do not distinguish these two, although the analysis tools that we propose allow for it. For example, one could collect human productions from one individual (*e.g.*, for personalisation) or from sub-populations (*e.g.* to improve fit for underrepresented communities).

**Other quality dimensions.** It is possible that a model fits various statistical properties of the human process (under $M_k(x)$, under $C_k(x)$, and for various choices of $k$) meanwhile none of its probable responses are humanly-accepted as a whole. This is why we shall think of our tools as statistical probes. We indeed find interesting instances that show good fit in terms of our distance probes but whose outputs may be perceived as inadequate. Manual inspection reveals that a marriage proposal in one of the dialogues (Figure 16 in the Appendix) is followed by a few incoherent model responses (*e.g.*., '*Thank you. It's not a question of the strength or weakness of the plot. I think it all falls within my capacity.*'), some dispreferred ones ('*If you want to have a hug?*'; see Levinson, 1983), and some with negative affect ('*I don't need your love. I know where you are coming from and I trust you will do the same.*'). Exhaustively defining all aspects of perceived quality (or human-likeness) is a strenuous endeavour which is highly dependent on the use case of the generation system. Our probes can be replaced with (possibly asymmetric) quality metrics which capture aspects (*e.g.*, affective content, toxicity, or readability) that are considered relevant for any given application.

## Acknowledgements

We thank Arabella Sinclair, Claire Gardent, Clara Meister, Christopher Lucas, Ehud Reiter, Sarenne Wallbridge, and the ILLC's Dialogue Modelling Group for inspiring discussions. We also thank the MaiNLP group for feedback on earlier drafts of this paper. MG and RF are supported by the European Research Council (ERC) under the European Union's Horizon 2020 research and innovation programme (grant agreement No. 819455). JB is supported by the ELLIS Amsterdam Unit. WA is supported by the EU's Horizon Europe research and innovation programme (grant agreement No. 101070631, UTTER). BP is supported by the European Research Council (ERC) (grant agreement No. 101043235).

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

# A  A Note on Entropy

Entropy is an information-theoretic concept that is often used to summarise uncertainty about a random variable. As useful as it may be in various contexts, entropy is not itself a complete characterisation of uncertainty (*e.g.*, two different distributions may have the same entropy, yet represent different uncertainty about their respective random variables). As we discuss in § 3.1, uncertainty about a random variable is fully represented by its underlying probability distribution (Halpern, 2017, Chapter 2).

Consider a discrete random variable $X$ with distribution $p_X$ and probability mass function (pmf) $p_X(x)$. Define the *surprisal* of an outcome $X = x$ as the quantity $-\log p_X(x)$. Then, *Shannon entropy* (or just *entropy* for short) is defined as the surprisal of $X$ taken in expectation under $p_X$ (MacKay, 2003). Due to the unbounded sample space and lack of conditional independence assumptions, entropy is intractable to compute for neural text generators. In some cases a Monte Carlo (MC) estimate of entropy can be formed with a reasonable amount of computation. For example, consider an autoregressive language model that assigns probability $f(x; \theta)$ to a complete sequence $x$ using a neural network with parameters $\theta$ (*e.g.*, an LSTM or Transformer). When we use ancestral sampling (Bishop, 2006) to decode from this model obtaining a sample $x^{(s)}$, the surprisal of $x^{(s)}$ is directly available via $-\log f(x; \theta)$, and the sample mean $-\frac{1}{S} \sum_{s=1}^{S} \log f(x^{(s)}; \theta)$ for $S$ ancestral samples forms an unbiased MC estimate for the entropy of $X$. In most cases, however, the

generator's pmf is unknown and the surprisal of an outcome is not available. That is the case, for example, whenever we employ decoding algorithms that bias away from the underlying distribution of the autoregressive LM—top-$p$, top-$k$, typical sampling are all good examples. The resulting pmf is then hard (or impossible) to characterise.

Furthermore, as much as Shannon entropy can be interpreted in its own information-theoretic terms, it is not immediately obvious how it can inform an analyst interested in the generator's faithfulness to human production variability. That said, the analyst may be interested in knowing, for example, that the entropy of the generator is similar to that of the 'human distribution' regardless of their ability to assign any useful interpretation to entropy proper. While we accept some analyst out there may be curious about that question, we refrain from performing such an analysis ourselves because (a) MC estimation is not available for most of the popular decoders we wanted to analyse, and (b) estimating the entropy of the human distribution requires a faithful model of it (that is, we need a perfectly faithful text generator to play the role of a 'gold standard').

## B  A Note on the Wasserstein 1-Distance

The Wasserstein 1-Distance $W_1(\cdot, \cdot)$ quantifies a notion of distance between two probability measures and is particularly convenient for it can be estimated using Dirac deltas (samples from those measures; Peyré et al., 2019) more easily than alternatives such as Kolmogorov–Smirnov and total variation distance (which require binning the measurements into empirical cdfs/pdfs). $W_1(M_k(x), H_k(x))$ and $W_1(C_k(x), H_k(x))$ have an interpretation in terms of 'mass' (in units of $k$) that has to be moved, on average, to transform one set of samples into another.

## C  Data Statistics

Table 1 shows relevant statistics for the four multiple-reference datasets presented in § 4.

## D  Additional Figures

Figure 7 shows human production variability over lexical and syntactic unigrams, bigrams, and trigrams (complementing Figure 2 in the main paper). Figure 8 shows the distribution of $\mu_{C_k(x)} - \mu_{H_k(x)}$ over instances for our four tasks (complementing Figure 3 in the main paper). Figures 9 to 11 show mean divergences across tasks, probes, and decoding algorithms (complementing Figure 5 in the main paper).

## E  Examples Discussed in the Qualitative Instance-Level Analysis

Figures 12-16 show examples of model fitness for the instances discussed in § 7.

## F  Decoding Configurations

Tables 2 to 5 show mean divergences for all the analysed decoding strategies, in terms of Wasserstein 1-Distance as well as mean distance.

|  |  | *Machine Translation* | | | *Text Simplification* | | | *Story Generation* | | | *Open-Domain Dialogue* | | |
|---|---|---|---|---|---|---|---|---|---|---|---|---|---|
|  |  | MEAN ± STD | MED. | RANGE | MEAN ± STD | MED. | RANGE | MEAN ± STD | MED. | RANGE | MEAN ± STD | MED. | RANGE |
| **Input** | Words | 23.34 ± 11.35 | 22 | 3-67 | 22.26 ± 8.92 | 21 | 7-57 | 25.40 ± 14.18 | 24 | 1-68 | 47.62 ± 30.37 | 40 | 5-311 |
|  | Tokens | 25.79 ± 12.91 | 23 | 4-81 | 28.00 ± 11.68 | 26 | 7-78 | 26.49 ± 14.68 | 24 | 1-70 | 48.94 ± 31.52 | 40 | 5-326 |
|  | Sentences | 1.01 ± 0.09 | 1 | 1-2 | 1.02 ± 0.14 | 1 | 1-2 | 1.75 ± 0.93 | 2 | 1-6 | 5.49 ± 2.82 | 5 | 1-22 |
|  | Words in sent. | 23.15 ± 11.37 | 22 | 2-67 | 21.80 ± 9.11 | 20 | 1-57 | 14.48 ± 7.89 | 14 | 1-50 | 8.67 ± 5.20 | 8 | 1-50 |
|  | Tokens in sent. | 25.58 ± 12.90 | 23 | 2-81 | 27.42 ± 11.88 | 25 | 1-78 | 15.12 ± 8.13 | 14 | 1-51 | 8.93 ± 5.39 | 8 | 1-50 |
| **Output** | Words | 21.96 ± 10.99 | 20 | 2-66 | 19.57 ± 8.29 | 18 | 4-62 | 659.72 ± 450.46 | 540 | 101-2681 | 10.61 ± 4.85 | 10 | 2-46 |
|  | Tokens | 27.28 ± 14.09 | 25 | 5-86 | 24.22 ± 10.65 | 22 | 5-91 | 696.66 ± 476.93 | 570 | 104-2961 | 10.84 ± 5.01 | 10 | 2-53 |
|  | Sentences | 1.06 ± 0.25 | 1 | 1-4 | 1.33 ± 0.56 | 1 | 1-5 | 47.76 ± 35.44 | 38 | 1-308 | 1.32 ± 0.52 | 1 | 1-5 |
|  | Words in sent. | 20.67 ± 10.86 | 19 | 1-66 | 14.70 ± 6.71 | 13 | 1-59 | 13.81 ± 9.59 | 12 | 1-722 | 8.06 ± 4.32 | 7 | 1-36 |
|  | Tokens in sent. | 25.69 ± 13.92 | 23 | 1-86 | 18.19 ± 8.78 | 16 | 1-91 | 14.63 ± 10.22 | 12 | 1-722 | 8.24 ± 4.45 | 8 | 2-37 |

Table 1: Length statistics. Number of tokens obtained with the tokenisers of the language models used for generation.

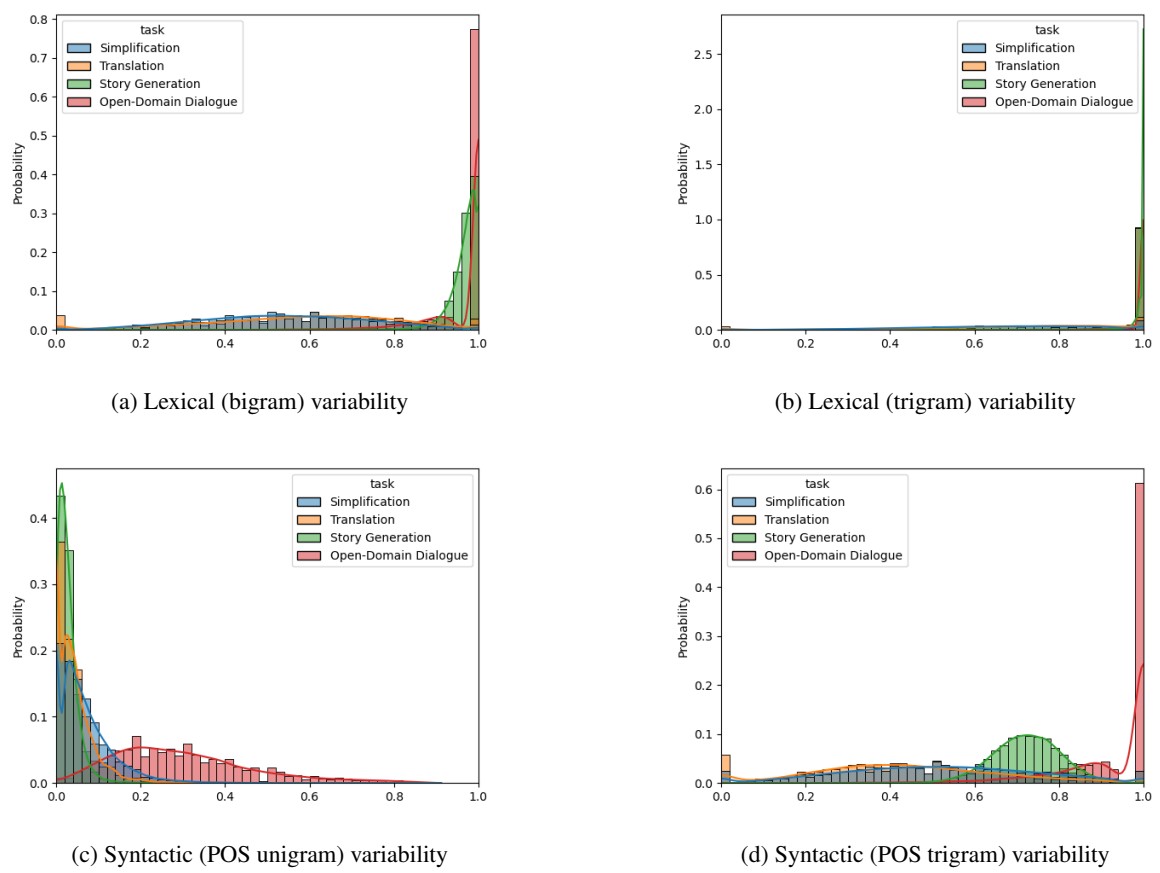

(a) Lexical (bigram) variability

(b) Lexical (trigram) variability

(c) Syntactic (POS unigram) variability

(d) Syntactic (POS trigram) variability

Figure 7: Human production variability across four NLG tasks (the remaining settings not reported in the main paper). The values on the $x$-axis are single samples of lexical or syntactic distance between two productions for each input (see Section 3). Probability mass on the right side signals high distance and thus high variability, and vice versa. A large spread indicates that production variability varies widely across inputs, and as such that a task does not define a specific level of variability.

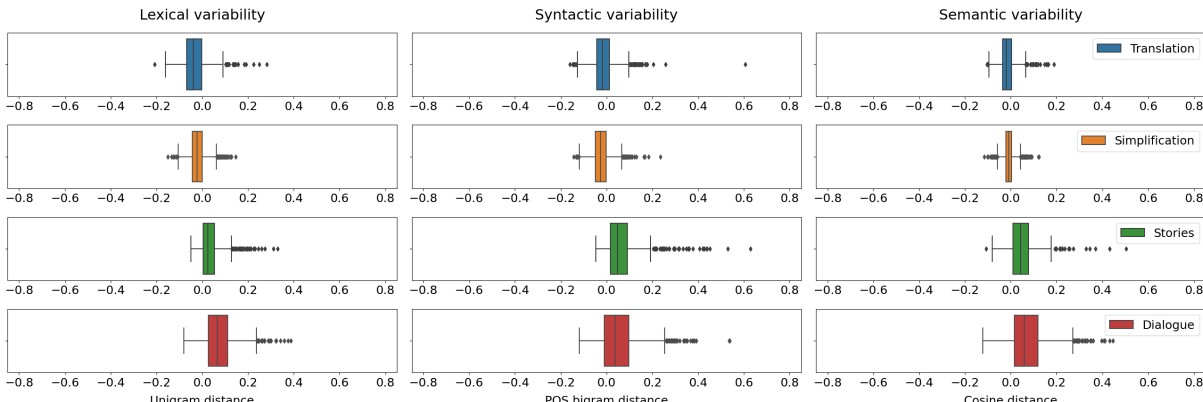

Figure 8: Distribution of $\mu_{C_k(x)} - \mu_{H_k(x)}$ over instances. Values greater than zero indicate the model overestimates the variability of the task (higher mean pairwise distance); values below zero indicate variability underestimation.

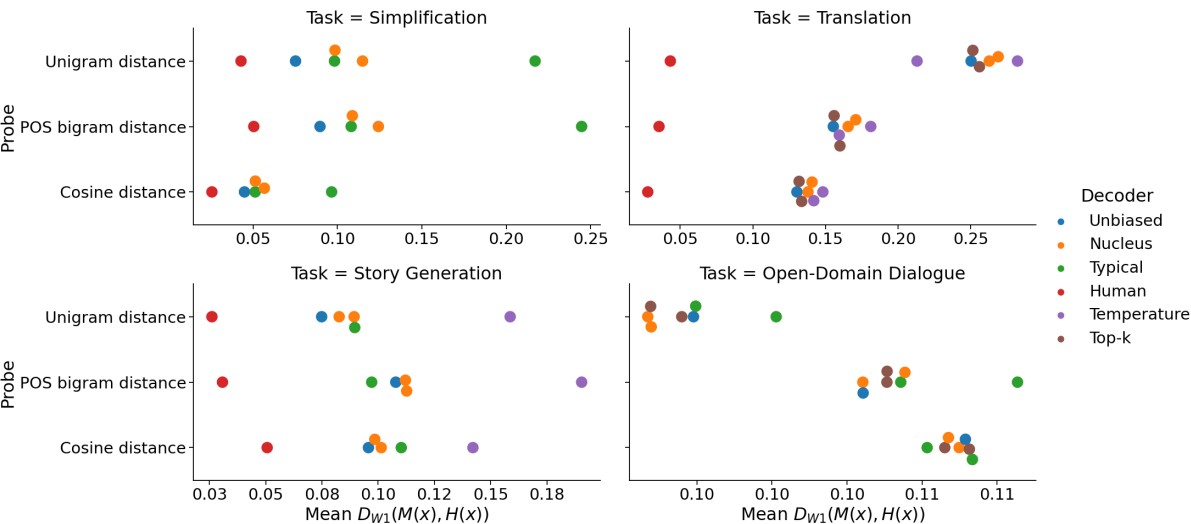

Figure 9: Mean Wasserstein distances $D_{W_1}(M(x), H(x))$ for (tasks, probe, decoding algorithm) tuples. Base models for each task are described in Section 4. Tuples that share colour have different decoding parameters. Human control group in red; except for dialogue, where 5 references are too few to create a control group.

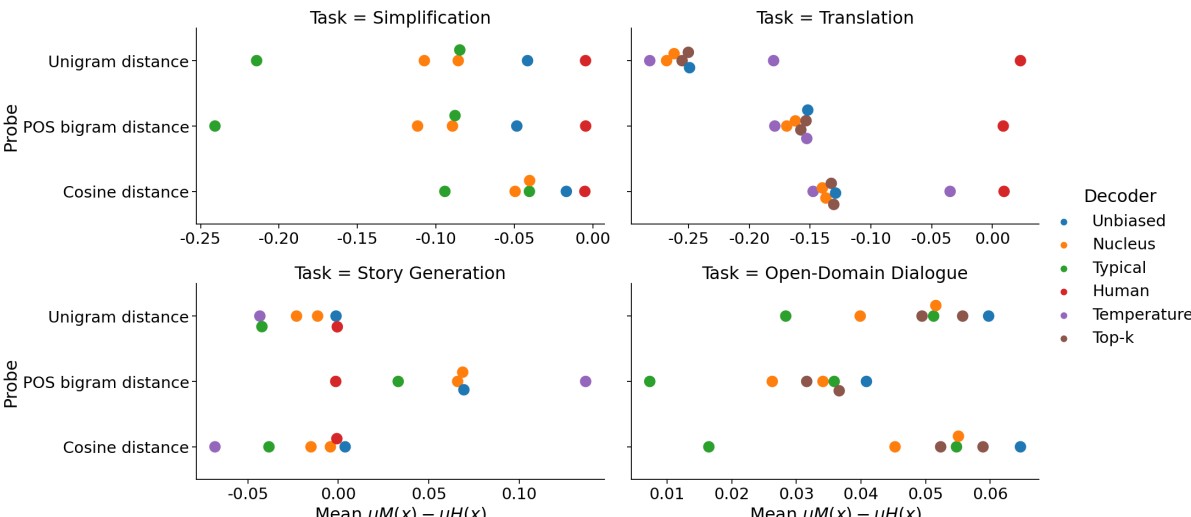

Figure 10: Mean of distances $\mu_{M(x)} - \mu_{H(x)}$ for (tasks, probe, decoding algorithm) tuples across test sets. Base models for each task are described in Section 4. Tuples that share colour have different decoding parameters. Human control group in red; except for dialogue, where 5 references are too few to create a control group.

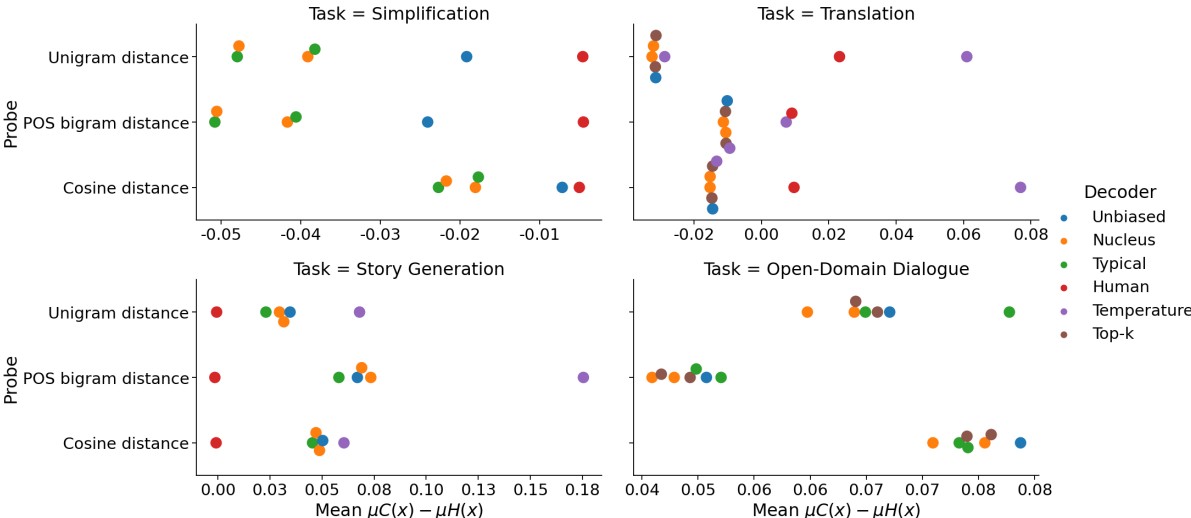

Figure 11: Mean of distances $\mu_{C(x)} - \mu_{H(x)}$ for (tasks, probe, decoding algorithm) tuples across test sets. Base models for each task are described in Section 4. Tuples that share colour have different decoding parameters. Human control group in red; except for dialogue, where 5 references are too few to create a control group.

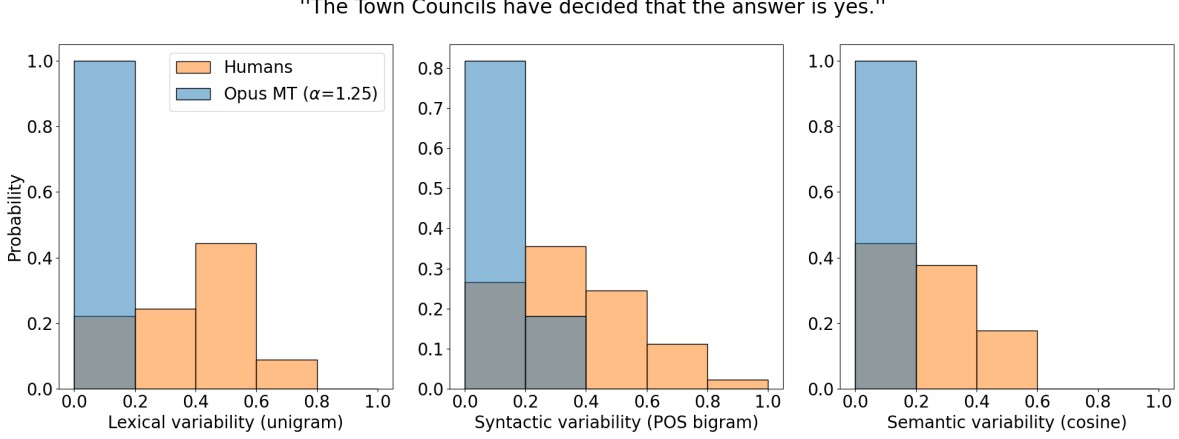

Figure 12: Example 1 of bad fitness $(D_{W_1}(M_k(x), H_k(x)))$ to human variability for the Opus MT model.

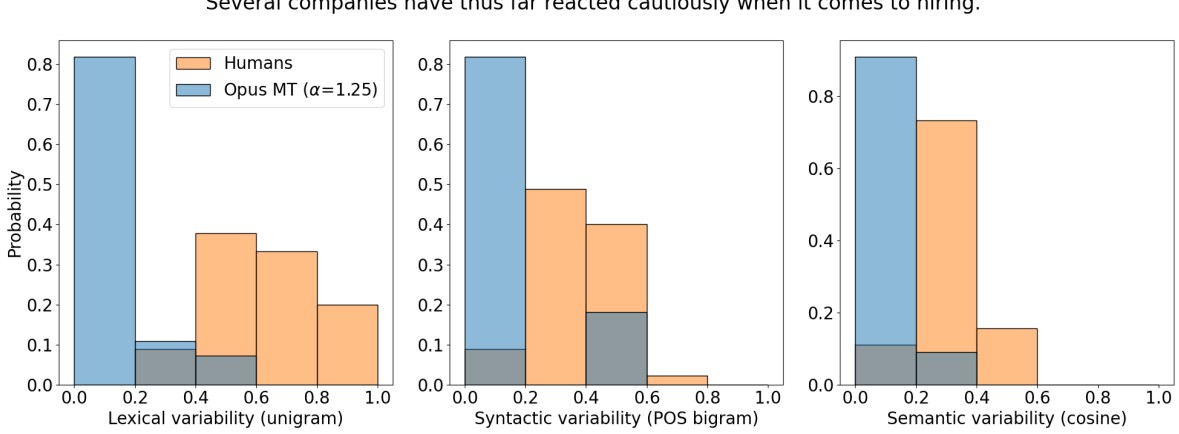

Figure 13: Example 2 of bad fitness $(D_{W_1}(M_k(x), H_k(x)))$ to human variability for the Opus MT model.

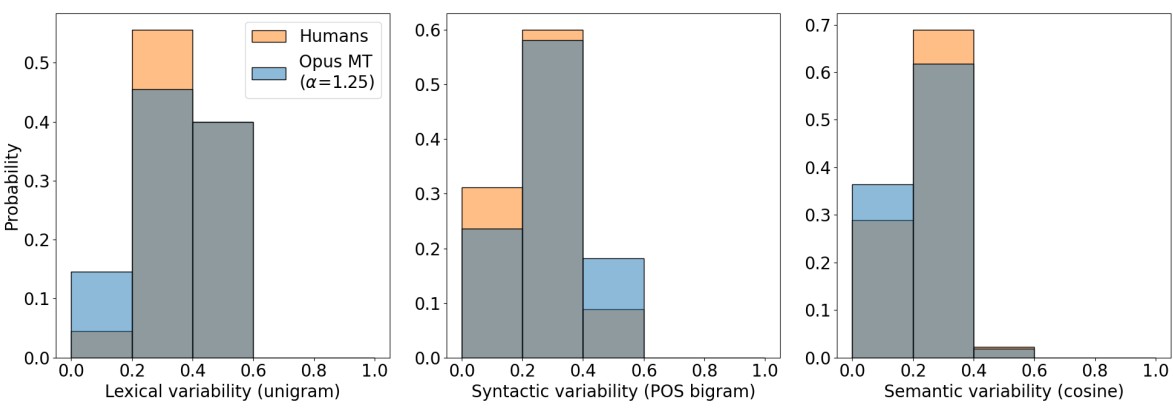

Figure 14: Example of good fitness ($D_{W_1}(M_k(x), H_k(x))$) to human variability for the Opus MT model.

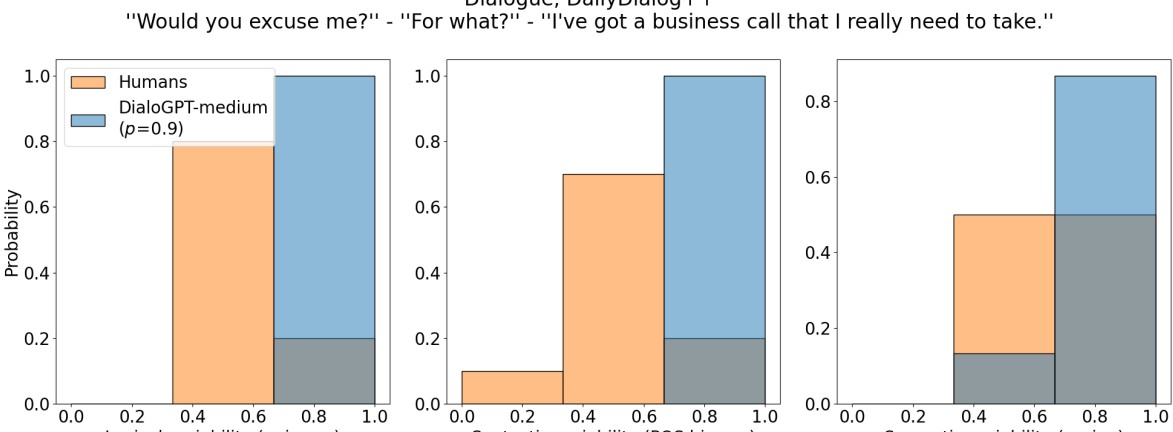

Figure 15: Example of bad fitness ($D_{W_1}(C_k(x), H_k(x))$) for DialoGPT-medium.

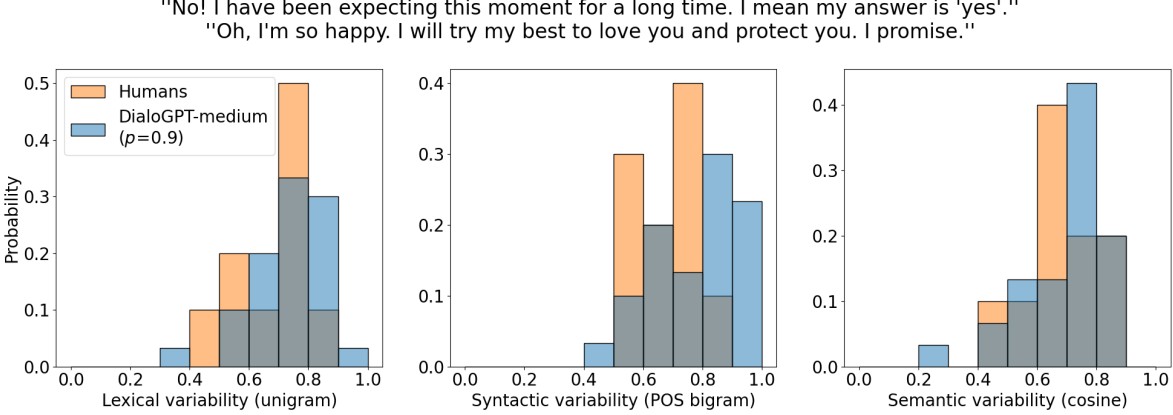

Figure 16: Example of good fitness ($D_{W_1}(C_k(x), H_k(x))$) for DialoGPT-medium.

Table 2: Mean $D_{W1}(M(x), H(x))$ results for different decoder settings.

| Model | Probe | Mean $D_{W1}(M(x), H(x))$ | Task |
|---|---|---|---|
| flanT5_large_finetuned-ancestral-val | Unigram distance | 0.075174 | Simplification |
| flanT5_large_finetuned-nucleus_09-val | Unigram distance | 0.114867 | Simplification |
| flanT5_large_finetuned-nucleus_095-val | Unigram distance | 0.098536 | Simplification |
| flanT5_large_finetuned-typical_02-val | Unigram distance | 0.217127 | Simplification |
| flanT5_large_finetuned-typical_095-val | Unigram distance | 0.098259 | Simplification |
| human_control | Unigram distance | 0.042863 | Simplification |
| flanT5_large_finetuned-ancestral-val | POS bigram distance | 0.089668 | Simplification |
| flanT5_large_finetuned-nucleus_09-val | POS bigram distance | 0.124259 | Simplification |
| flanT5_large_finetuned-nucleus_095-val | POS bigram distance | 0.108829 | Simplification |
| flanT5_large_finetuned-typical_02-val | POS bigram distance | 0.244697 | Simplification |
| flanT5_large_finetuned-typical_095-val | POS bigram distance | 0.108139 | Simplification |
| human_control | POS bigram distance | 0.050427 | Simplification |
| flanT5_large_finetuned-ancestral-val | Cosine distance | 0.044924 | Simplification |
| flanT5_large_finetuned-nucleus_09-val | Cosine distance | 0.056711 | Simplification |
| flanT5_large_finetuned-nucleus_095-val | Cosine distance | 0.051341 | Simplification |
| flanT5_large_finetuned-typical_02-val | Cosine distance | 0.096548 | Simplification |
| flanT5_large_finetuned-typical_095-val | Cosine distance | 0.051276 | Simplification |
| human_control | Cosine distance | 0.025631 | Simplification |
| opus-ancestral | Unigram distance | 0.250246 | Translation |
| opus-nucleus_085 | Unigram distance | 0.268903 | Translation |
| opus-nucleus_09 | Unigram distance | 0.262719 | Translation |
| opus-temperature05 | Unigram distance | 0.213019 | Translation |
| opus-temperature075 | Unigram distance | 0.282174 | Translation |
| opus-top_k_30 | Unigram distance | 0.255974 | Translation |
| opus-top_k_40 | Unigram distance | 0.251440 | Translation |
| human_control | Unigram distance | 0.043193 | Translation |
| opus-ancestral | POS bigram distance | 0.155366 | Translation |
| opus-nucleus_085 | POS bigram distance | 0.170862 | Translation |
| opus-nucleus_09 | POS bigram distance | 0.165579 | Translation |
| opus-temperature05 | POS bigram distance | 0.159474 | Translation |
| opus-temperature075 | POS bigram distance | 0.181173 | Translation |
| opus-top_k_30 | POS bigram distance | 0.159928 | Translation |
| opus-top_k_40 | POS bigram distance | 0.155916 | Translation |
| human_control | POS bigram distance | 0.035402 | Translation |
| opus-ancestral | Cosine distance | 0.130312 | Translation |
| opus-nucleus_085 | Cosine distance | 0.140748 | Translation |
| opus-nucleus_09 | Cosine distance | 0.137961 | Translation |
| opus-temperature05 | Cosine distance | 0.141920 | Translation |
| opus-temperature075 | Cosine distance | 0.148178 | Translation |
| opus-top_k_30 | Cosine distance | 0.133588 | Translation |
| opus-top_k_40 | Cosine distance | 0.131818 | Translation |
| human_control | Cosine distance | 0.027646 | Translation |
| gpt2_large_finetuned-ancestral-test | Unigram distance | 0.074878 | Story Generation |
| gpt2_large_finetuned-nucleus_09-test | Unigram distance | 0.089306 | Story Generation |
| gpt2_large_finetuned-nucleus_095-test | Unigram distance | 0.082663 | Story Generation |
| gpt2_large_finetuned-temperature05-test | Unigram distance | 0.158670 | Story Generation |
| gpt2_large_finetuned-typical_02-test | Unigram distance | 0.089595 | Story Generation |
| human_control | Unigram distance | 0.026115 | Story Generation |
| gpt2_large_finetuned-ancestral-test | POS bigram distance | 0.107836 | Story Generation |
| gpt2_large_finetuned-nucleus_09-test | POS bigram distance | 0.112113 | Story Generation |
| gpt2_large_finetuned-nucleus_095-test | POS bigram distance | 0.112662 | Story Generation |
| gpt2_large_finetuned-temperature05-test | POS bigram distance | 0.190448 | Story Generation |
| gpt2_large_finetuned-typical_02-test | POS bigram distance | 0.097114 | Story Generation |
| human_control | POS bigram distance | 0.030801 | Story Generation |
| gpt2_large_finetuned-ancestral-test | Cosine distance | 0.095689 | Story Generation |
| gpt2_large_finetuned-nucleus_09-test | Cosine distance | 0.101362 | Story Generation |
| gpt2_large_finetuned-nucleus_095-test | Cosine distance | 0.098502 | Story Generation |
| gpt2_large_finetuned-temperature05-test | Cosine distance | 0.142114 | Story Generation |
| gpt2_large_finetuned-typical_02-test | Cosine distance | 0.110264 | Story Generation |
| human_control | Cosine distance | 0.050663 | Story Generation |
| dialogpt_large-ancestral-dev | Unigram distance | 0.094768 | Open-Domain Dialogue |
| human_control | Unigram distance | NaN | Open-Domain Dialogue |
| dialogpt_large-nucleus_09-dev | Unigram distance | 0.091718 | Open-Domain Dialogue |
| dialogpt_large-nucleus_095-dev | Unigram distance | 0.091948 | Open-Domain Dialogue |
| dialogpt_large-top_k_30-dev | Unigram distance | 0.091910 | Open-Domain Dialogue |
| dialogpt_large-top_k_40-dev | Unigram distance | 0.093984 | Open-Domain Dialogue |
| dialogpt_large-typical_02-dev | Unigram distance | 0.100279 | Open-Domain Dialogue |
| dialogpt_large-typical_095-dev | Unigram distance | 0.094912 | Open-Domain Dialogue |
| dialogpt_large-ancestral-dev | POS bigram distance | 0.106077 | Open-Domain Dialogue |
| human_control | POS bigram distance | NaN | Open-Domain Dialogue |
| dialogpt_large-nucleus_09-dev | POS bigram distance | 0.108866 | Open-Domain Dialogue |
| dialogpt_large-nucleus_095-dev | POS bigram distance | 0.106057 | Open-Domain Dialogue |
| dialogpt_large-top_k_30-dev | POS bigram distance | 0.107674 | Open-Domain Dialogue |
| dialogpt_large-top_k_40-dev | POS bigram distance | 0.107663 | Open-Domain Dialogue |
| dialogpt_large-typical_02-dev | POS bigram distance | 0.116369 | Open-Domain Dialogue |
| dialogpt_large-typical_095-dev | POS bigram distance | 0.108587 | Open-Domain Dialogue |
| dialogpt_large-ancestral-dev | Cosine distance | 0.112896 | Open-Domain Dialogue |
| human_control | Cosine distance | NaN | Open-Domain Dialogue |
| dialogpt_large-nucleus_09-dev | Cosine distance | 0.112480 | Open-Domain Dialogue |
| dialogpt_large-nucleus_095-dev | Cosine distance | 0.111765 | Open-Domain Dialogue |
| dialogpt_large-top_k_30-dev | Cosine distance | 0.111521 | Open-Domain Dialogue |
| dialogpt_large-top_k_40-dev | Cosine distance | 0.113160 | Open-Domain Dialogue |
| dialogpt_large-typical_02-dev | Cosine distance | 0.113362 | Open-Domain Dialogue |
| dialogpt_large-typical_095-dev | Cosine distance | 0.110345 | Open-Domain Dialogue |

Table 3: Mean $D_\mu(M(x), H(x))$ for different decoder settings

| Model | Probe | Mean $D_\mu(M(x), H(x))$ | Task |
|---|---|---|---|
| flanT5_large_finetuned-ancestral-val | Unigram distance | -0.041574 | Simplification |
| flanT5_large_finetuned-nucleus_09-val | Unigram distance | -0.107347 | Simplification |
| flanT5_large_finetuned-nucleus_095-val | Unigram distance | -0.085751 | Simplification |
| flanT5_large_finetuned-typical_02-val | Unigram distance | -0.214236 | Simplification |
| flanT5_large_finetuned-typical_095-val | Unigram distance | -0.084748 | Simplification |
| human_control | Unigram distance | -0.004613 | Simplification |
| flanT5_large_finetuned-ancestral-val | POS bigram distance | -0.048417 | Simplification |
| flanT5_large_finetuned-nucleus_09-val | POS bigram distance | -0.111659 | Simplification |
| flanT5_large_finetuned-nucleus_095-val | POS bigram distance | -0.089422 | Simplification |
| flanT5_large_finetuned-typical_02-val | POS bigram distance | -0.240776 | Simplification |
| flanT5_large_finetuned-typical_095-val | POS bigram distance | -0.087786 | Simplification |
| human_control | POS bigram distance | -0.004535 | Simplification |
| flanT5_large_finetuned-ancestral-val | Cosine distance | -0.016843 | Simplification |
| flanT5_large_finetuned-nucleus_09-val | Cosine distance | -0.049492 | Simplification |
| flanT5_large_finetuned-nucleus_095-val | Cosine distance | -0.040255 | Simplification |
| flanT5_large_finetuned-typical_02-val | Cosine distance | -0.094252 | Simplification |
| flanT5_large_finetuned-typical_095-val | Cosine distance | -0.040386 | Simplification |
| human_control | Cosine distance | -0.005045 | Simplification |
| opus-ancestral | Unigram distance | -0.249243 | Translation |
| opus-nucleus_085 | Unigram distance | -0.268126 | Translation |
| opus-nucleus_09 | Unigram distance | -0.261954 | Translation |
| opus-temperature05 | Unigram distance | -0.180041 | Translation |
| opus-temperature075 | Unigram distance | -0.281996 | Translation |
| opus-top_k_30 | Unigram distance | -0.255388 | Translation |
| opus-top_k_40 | Unigram distance | -0.250243 | Translation |
| human_control | Unigram distance | 0.023181 | Translation |
| opus-ancestral | POS bigram distance | -0.151919 | Translation |
| opus-nucleus_085 | POS bigram distance | -0.169243 | Translation |
| opus-nucleus_09 | POS bigram distance | -0.162130 | Translation |
| opus-temperature05 | POS bigram distance | -0.152725 | Translation |
| opus-temperature075 | POS bigram distance | -0.179065 | Translation |
| opus-top_k_30 | POS bigram distance | -0.157629 | Translation |
| opus-top_k_40 | POS bigram distance | -0.153329 | Translation |
| human_control | POS bigram distance | 0.009030 | Translation |
| opus-ancestral | Cosine distance | -0.129037 | Translation |
| opus-nucleus_085 | Cosine distance | -0.139889 | Translation |
| opus-nucleus_09 | Cosine distance | -0.136995 | Translation |
| opus-temperature05 | Cosine distance | -0.034734 | Translation |
| opus-temperature075 | Cosine distance | -0.147533 | Translation |
| opus-top_k_30 | Cosine distance | -0.132567 | Translation |
| opus-top_k_40 | Cosine distance | -0.130491 | Translation |
| human_control | Cosine distance | 0.009674 | Translation |
| gpt2_large_finetuned-ancestral-test | Unigram distance | -0.001277 | Story Generation |
| gpt2_large_finetuned-nucleus_09-test | Unigram distance | -0.023124 | Story Generation |
| gpt2_large_finetuned-nucleus_095-test | Unigram distance | -0.011460 | Story Generation |
| gpt2_large_finetuned-temperature05-test | Unigram distance | -0.043398 | Story Generation |
| gpt2_large_finetuned-typical_02-test | Unigram distance | -0.042259 | Story Generation |
| human_control | Unigram distance | -0.000588 | Story Generation |
| gpt2_large_finetuned-ancestral-test | POS bigram distance | 0.069409 | Story Generation |
| gpt2_large_finetuned-nucleus_09-test | POS bigram distance | 0.065952 | Story Generation |
| gpt2_large_finetuned-nucleus_095-test | POS bigram distance | 0.068751 | Story Generation |
| gpt2_large_finetuned-temperature05-test | POS bigram distance | 0.136734 | Story Generation |
| gpt2_large_finetuned-typical_02-test | POS bigram distance | 0.033082 | Story Generation |
| human_control | POS bigram distance | -0.001439 | Story Generation |
| gpt2_large_finetuned-ancestral-test | Cosine distance | 0.003763 | Story Generation |
| gpt2_large_finetuned-nucleus_09-test | Cosine distance | -0.015119 | Story Generation |
| gpt2_large_finetuned-nucleus_095-test | Cosine distance | -0.004399 | Story Generation |
| gpt2_large_finetuned-temperature05-test | Cosine distance | -0.068208 | Story Generation |
| gpt2_large_finetuned-typical_02-test | Cosine distance | -0.038362 | Story Generation |
| human_control | Cosine distance | -0.000839 | Story Generation |
| dialogpt_large-ancestral-dev | Unigram distance | 0.059756 | Open-Domain Dialogue |
| human_control | Unigram distance | NaN | Open-Domain Dialogue |
| dialogpt_large-nucleus_09-dev | Unigram distance | 0.039885 | Open-Domain Dialogue |
| dialogpt_large-nucleus_095-dev | Unigram distance | 0.051588 | Open-Domain Dialogue |
| dialogpt_large-top_k_30-dev | Unigram distance | 0.049452 | Open-Domain Dialogue |
| dialogpt_large-top_k_40-dev | Unigram distance | 0.055738 | Open-Domain Dialogue |
| dialogpt_large-typical_02-dev | Unigram distance | 0.028369 | Open-Domain Dialogue |
| dialogpt_large-typical_095-dev | Unigram distance | 0.051247 | Open-Domain Dialogue |
| dialogpt_large-ancestral-dev | POS bigram distance | 0.040850 | Open-Domain Dialogue |
| human_control | POS bigram distance | NaN | Open-Domain Dialogue |
| dialogpt_large-nucleus_09-dev | POS bigram distance | 0.026282 | Open-Domain Dialogue |
| dialogpt_large-nucleus_095-dev | POS bigram distance | 0.034128 | Open-Domain Dialogue |
| dialogpt_large-top_k_30-dev | POS bigram distance | 0.031613 | Open-Domain Dialogue |
| dialogpt_large-top_k_40-dev | POS bigram distance | 0.036636 | Open-Domain Dialogue |
| dialogpt_large-typical_02-dev | POS bigram distance | 0.007331 | Open-Domain Dialogue |
| dialogpt_large-typical_095-dev | POS bigram distance | 0.035866 | Open-Domain Dialogue |
| dialogpt_large-ancestral-dev | Cosine distance | 0.064680 | Open-Domain Dialogue |
| human_control | Cosine distance | NaN | Open-Domain Dialogue |
| dialogpt_large-nucleus_09-dev | Cosine distance | 0.045279 | Open-Domain Dialogue |
| dialogpt_large-nucleus_095-dev | Cosine distance | 0.055085 | Open-Domain Dialogue |
| dialogpt_large-top_k_30-dev | Cosine distance | 0.052337 | Open-Domain Dialogue |
| dialogpt_large-top_k_40-dev | Cosine distance | 0.058887 | Open-Domain Dialogue |
| dialogpt_large-typical_02-dev | Cosine distance | 0.016471 | Open-Domain Dialogue |
| dialogpt_large-typical_095-dev | Cosine distance | 0.054787 | Open-Domain Dialogue |

Table 4: Mean $D_{W1}(C(x), H(x))$ results for multiple decoding settings.

| Model | Probe | Mean $D_{W1}(C(x), H(x))$ | Task |
|---|---|---|---|
| flanT5_large_finetuned-ancestral-val | Unigram distance | 0.047961 | Simplification |
| flanT5_large_finetuned-nucleus_09-val | Unigram distance | 0.059546 | Simplification |
| flanT5_large_finetuned-nucleus_095-val | Unigram distance | 0.054702 | Simplification |
| flanT5_large_finetuned-typical_02-val | Unigram distance | 0.068984 | Simplification |
| flanT5_large_finetuned-typical_095-val | Unigram distance | 0.054952 | Simplification |
| human_control | Unigram distance | 0.042863 | Simplification |
| flanT5_large_finetuned-ancestral-val | POS bigram distance | 0.056443 | Simplification |
| flanT5_large_finetuned-nucleus_09-val | POS bigram distance | 0.065849 | Simplification |
| flanT5_large_finetuned-nucleus_095-val | POS bigram distance | 0.061433 | Simplification |
| flanT5_large_finetuned-typical_02-val | POS bigram distance | 0.078297 | Simplification |
| flanT5_large_finetuned-typical_095-val | POS bigram distance | 0.061684 | Simplification |
| human_control | POS bigram distance | 0.050427 | Simplification |
| flanT5_large_finetuned-ancestral-val | Cosine distance | 0.027983 | Simplification |
| flanT5_large_finetuned-nucleus_09-val | Cosine distance | 0.030626 | Simplification |
| flanT5_large_finetuned-nucleus_095-val | Cosine distance | 0.029443 | Simplification |
| flanT5_large_finetuned-typical_02-val | Cosine distance | 0.036008 | Simplification |
| flanT5_large_finetuned-typical_095-val | Cosine distance | 0.029270 | Simplification |
| human_control | Cosine distance | 0.025631 | Simplification |
| opus-ancestral | Unigram distance | 0.067614 | Translation |
| opus-nucleus_085 | Unigram distance | 0.069170 | Translation |
| opus-nucleus_09 | Unigram distance | 0.068676 | Translation |
| opus-temperature05 | Unigram distance | 0.125683 | Translation |
| opus-temperature075 | Unigram distance | 0.070435 | Translation |
| opus-top_k_30 | Unigram distance | 0.068002 | Translation |
| opus-top_k_40 | Unigram distance | 0.066907 | Translation |
| human_control | Unigram distance | 0.043193 | Translation |
| opus-ancestral | POS bigram distance | 0.055636 | Translation |
| opus-nucleus_085 | POS bigram distance | 0.057639 | Translation |
| opus-nucleus_09 | POS bigram distance | 0.056588 | Translation |
| opus-temperature05 | POS bigram distance | 0.064878 | Translation |
| opus-temperature075 | POS bigram distance | 0.058195 | Translation |
| opus-top_k_30 | POS bigram distance | 0.056383 | Translation |
| opus-top_k_40 | POS bigram distance | 0.055540 | Translation |
| human_control | POS bigram distance | 0.035402 | Translation |
| opus-ancestral | Cosine distance | 0.043123 | Translation |
| opus-nucleus_085 | Cosine distance | 0.043951 | Translation |
| opus-nucleus_09 | Cosine distance | 0.043713 | Translation |
| opus-temperature05 | Cosine distance | 0.112590 | Translation |
| opus-temperature075 | Cosine distance | 0.045421 | Translation |
| opus-top_k_30 | Cosine distance | 0.043264 | Translation |
| opus-top_k_40 | Cosine distance | 0.042813 | Translation |
| human_control | Cosine distance | 0.027646 | Translation |
| gpt2_large_finetuned-ancestral-test | Unigram distance | 0.052815 | Story Generation |
| gpt2_large_finetuned-nucleus_09-test | Unigram distance | 0.052926 | Story Generation |
| gpt2_large_finetuned-nucleus_095-test | Unigram distance | 0.053015 | Story Generation |
| gpt2_large_finetuned-temperature05-test | Unigram distance | 0.087166 | Story Generation |
| gpt2_large_finetuned-typical_02-test | Unigram distance | 0.050375 | Story Generation |
| human_control | Unigram distance | 0.026115 | Story Generation |
| gpt2_large_finetuned-ancestral-test | POS bigram distance | 0.082493 | Story Generation |
| gpt2_large_finetuned-nucleus_09-test | POS bigram distance | 0.088013 | Story Generation |
| gpt2_large_finetuned-nucleus_095-test | POS bigram distance | 0.085413 | Story Generation |
| gpt2_large_finetuned-temperature05-test | POS bigram distance | 0.180557 | Story Generation |
| gpt2_large_finetuned-typical_02-test | POS bigram distance | 0.078543 | Story Generation |
| human_control | POS bigram distance | 0.030801 | Story Generation |
| gpt2_large_finetuned-ancestral-test | Cosine distance | 0.078816 | Story Generation |
| gpt2_large_finetuned-nucleus_09-test | Cosine distance | 0.077738 | Story Generation |
| gpt2_large_finetuned-nucleus_095-test | Cosine distance | 0.078084 | Story Generation |
| gpt2_large_finetuned-temperature05-test | Cosine distance | 0.087292 | Story Generation |
| gpt2_large_finetuned-typical_02-test | Cosine distance | 0.078043 | Story Generation |
| human_control | Cosine distance | 0.050663 | Story Generation |
| dialogpt_large-ancestral-dev | Unigram distance | 0.082540 | Open-Domain Dialogue |
| human_control | Unigram distance | NaN | Open-Domain Dialogue |
| dialogpt_large-nucleus_09-dev | Unigram distance | 0.078856 | Open-Domain Dialogue |
| dialogpt_large-nucleus_095-dev | Unigram distance | 0.080944 | Open-Domain Dialogue |
| dialogpt_large-top_k_30-dev | Unigram distance | 0.081368 | Open-Domain Dialogue |
| dialogpt_large-top_k_40-dev | Unigram distance | 0.082693 | Open-Domain Dialogue |
| dialogpt_large-typical_02-dev | Unigram distance | 0.090923 | Open-Domain Dialogue |
| dialogpt_large-typical_095-dev | Unigram distance | 0.081366 | Open-Domain Dialogue |
| dialogpt_large-ancestral-dev | POS bigram distance | 0.085033 | Open-Domain Dialogue |
| human_control | POS bigram distance | NaN | Open-Domain Dialogue |
| dialogpt_large-nucleus_09-dev | POS bigram distance | 0.085009 | Open-Domain Dialogue |
| dialogpt_large-nucleus_095-dev | POS bigram distance | 0.084188 | Open-Domain Dialogue |
| dialogpt_large-top_k_30-dev | POS bigram distance | 0.084248 | Open-Domain Dialogue |
| dialogpt_large-top_k_40-dev | POS bigram distance | 0.083804 | Open-Domain Dialogue |
| dialogpt_large-typical_02-dev | POS bigram distance | 0.088119 | Open-Domain Dialogue |
| dialogpt_large-typical_095-dev | POS bigram distance | 0.085461 | Open-Domain Dialogue |
| dialogpt_large-ancestral-dev | Cosine distance | 0.097410 | Open-Domain Dialogue |
| human_control | Cosine distance | NaN | Open-Domain Dialogue |
| dialogpt_large-nucleus_09-dev | Cosine distance | 0.094412 | Open-Domain Dialogue |
| dialogpt_large-nucleus_095-dev | Cosine distance | 0.096555 | Open-Domain Dialogue |
| dialogpt_large-top_k_30-dev | Cosine distance | 0.095628 | Open-Domain Dialogue |
| dialogpt_large-top_k_40-dev | Cosine distance | 0.096752 | Open-Domain Dialogue |
| dialogpt_large-typical_02-dev | Cosine distance | 0.095108 | Open-Domain Dialogue |
| dialogpt_large-typical_095-dev | Cosine distance | 0.095468 | Open-Domain Dialogue |

Table 5: Mean $D_\mu(C(x), H(x)))$ results for different decoder parameter settings.

| Model | Probe | Mean $D_\mu(C(x), H(x)))$ | Task |
|---|---|---|---|
| flanT5_large_finetuned-ancestral-val | Unigram distance | -0.019180 | Simplification |
| flanT5_large_finetuned-nucleus_09-val | Unigram distance | -0.047746 | Simplification |
| flanT5_large_finetuned-nucleus_095-val | Unigram distance | -0.039108 | Simplification |
| flanT5_large_finetuned-typical_02-val | Unigram distance | -0.047960 | Simplification |
| flanT5_large_finetuned-typical_095-val | Unigram distance | -0.038222 | Simplification |
| human_control | Unigram distance | -0.004613 | Simplification |
| flanT5_large_finetuned-ancestral-val | POS bigram distance | -0.024071 | Simplification |
| flanT5_large_finetuned-nucleus_09-val | POS bigram distance | -0.050534 | Simplification |
| flanT5_large_finetuned-nucleus_095-val | POS bigram distance | -0.041670 | Simplification |
| flanT5_large_finetuned-typical_02-val | POS bigram distance | -0.050770 | Simplification |
| flanT5_large_finetuned-typical_095-val | POS bigram distance | -0.040591 | Simplification |
| human_control | POS bigram distance | -0.004535 | Simplification |
| flanT5_large_finetuned-ancestral-val | Cosine distance | -0.007191 | Simplification |
| flanT5_large_finetuned-nucleus_09-val | Cosine distance | -0.021733 | Simplification |
| flanT5_large_finetuned-nucleus_095-val | Cosine distance | -0.018084 | Simplification |
| flanT5_large_finetuned-typical_02-val | Cosine distance | -0.022718 | Simplification |
| flanT5_large_finetuned-typical_095-val | Cosine distance | -0.017714 | Simplification |
| human_control | Cosine distance | -0.005045 | Simplification |
| opus-ancestral | Unigram distance | -0.031401 | Translation |
| opus-nucleus_085 | Unigram distance | -0.032499 | Translation |
| opus-nucleus_09 | Unigram distance | -0.032062 | Translation |
| opus-temperature05 | Unigram distance | 0.060986 | Translation |
| opus-temperature075 | Unigram distance | -0.028734 | Translation |
| opus-top_k_30 | Unigram distance | -0.031515 | Translation |
| opus-top_k_40 | Unigram distance | -0.031321 | Translation |
| human_control | Unigram distance | 0.023181 | Translation |
| opus-ancestral | POS bigram distance | -0.010164 | Translation |
| opus-nucleus_085 | POS bigram distance | -0.011306 | Translation |
| opus-nucleus_09 | POS bigram distance | -0.010588 | Translation |
| opus-temperature05 | POS bigram distance | 0.007351 | Translation |
| opus-temperature075 | POS bigram distance | -0.009418 | Translation |
| opus-top_k_30 | POS bigram distance | -0.010525 | Translation |
| opus-top_k_40 | POS bigram distance | -0.010689 | Translation |
| human_control | POS bigram distance | 0.009030 | Translation |
| opus-ancestral | Cosine distance | -0.014492 | Translation |
| opus-nucleus_085 | Cosine distance | -0.015246 | Translation |
| opus-nucleus_09 | Cosine distance | -0.015266 | Translation |
| opus-temperature05 | Cosine distance | 0.076953 | Translation |
| opus-temperature075 | Cosine distance | -0.013291 | Translation |
| opus-top_k_30 | Cosine distance | -0.014714 | Translation |
| opus-top_k_40 | Cosine distance | -0.014507 | Translation |
| human_control | Cosine distance | 0.009674 | Translation |
| gpt2_large_finetuned-ancestral-test | Unigram distance | 0.034652 | Story Generation |
| gpt2_large_finetuned-nucleus_09-test | Unigram distance | 0.029568 | Story Generation |
| gpt2_large_finetuned-nucleus_095-test | Unigram distance | 0.031654 | Story Generation |
| gpt2_large_finetuned-temperature05-test | Unigram distance | 0.068008 | Story Generation |
| gpt2_large_finetuned-typical_02-test | Unigram distance | 0.023071 | Story Generation |
| human_control | Unigram distance | -0.000588 | Story Generation |
| gpt2_large_finetuned-ancestral-test | POS bigram distance | 0.066966 | Story Generation |
| gpt2_large_finetuned-nucleus_09-test | POS bigram distance | 0.073412 | Story Generation |
| gpt2_large_finetuned-nucleus_095-test | POS bigram distance | 0.069038 | Story Generation |
| gpt2_large_finetuned-temperature05-test | POS bigram distance | 0.175454 | Story Generation |
| gpt2_large_finetuned-typical_02-test | POS bigram distance | 0.058097 | Story Generation |
| human_control | POS bigram distance | -0.001439 | Story Generation |
| gpt2_large_finetuned-ancestral-test | Cosine distance | 0.050337 | Story Generation |
| gpt2_large_finetuned-nucleus_09-test | Cosine distance | 0.047109 | Story Generation |
| gpt2_large_finetuned-nucleus_095-test | Cosine distance | 0.048778 | Story Generation |
| gpt2_large_finetuned-temperature05-test | Cosine distance | 0.060529 | Story Generation |
| gpt2_large_finetuned-typical_02-test | Cosine distance | 0.045444 | Story Generation |
| human_control | Cosine distance | -0.000839 | Story Generation |
| dialogpt_large-ancestral-dev | Unigram distance | 0.067089 | Open-Domain Dialogue |
| human_control | Unigram distance | NaN | Open-Domain Dialogue |
| dialogpt_large-nucleus_09-dev | Unigram distance | 0.059751 | Open-Domain Dialogue |
| dialogpt_large-nucleus_095-dev | Unigram distance | 0.063929 | Open-Domain Dialogue |
| dialogpt_large-top_k_30-dev | Unigram distance | 0.064058 | Open-Domain Dialogue |
| dialogpt_large-top_k_40-dev | Unigram distance | 0.065997 | Open-Domain Dialogue |
| dialogpt_large-typical_02-dev | Unigram distance | 0.077755 | Open-Domain Dialogue |
| dialogpt_large-typical_095-dev | Unigram distance | 0.064922 | Open-Domain Dialogue |
| dialogpt_large-ancestral-dev | POS bigram distance | 0.050752 | Open-Domain Dialogue |
| human_control | POS bigram distance | NaN | Open-Domain Dialogue |
| dialogpt_large-nucleus_09-dev | POS bigram distance | 0.045903 | Open-Domain Dialogue |
| dialogpt_large-nucleus_095-dev | POS bigram distance | 0.047880 | Open-Domain Dialogue |
| dialogpt_large-top_k_30-dev | POS bigram distance | 0.046732 | Open-Domain Dialogue |
| dialogpt_large-top_k_40-dev | POS bigram distance | 0.049299 | Open-Domain Dialogue |
| dialogpt_large-typical_02-dev | POS bigram distance | 0.052058 | Open-Domain Dialogue |
| dialogpt_large-typical_095-dev | POS bigram distance | 0.049841 | Open-Domain Dialogue |
| dialogpt_large-ancestral-dev | Cosine distance | 0.078752 | Open-Domain Dialogue |
| human_control | Cosine distance | NaN | Open-Domain Dialogue |
| dialogpt_large-nucleus_09-dev | Cosine distance | 0.070942 | Open-Domain Dialogue |
| dialogpt_large-nucleus_095-dev | Cosine distance | 0.075569 | Open-Domain Dialogue |
| dialogpt_large-top_k_30-dev | Cosine distance | 0.073973 | Open-Domain Dialogue |
| dialogpt_large-top_k_40-dev | Cosine distance | 0.076133 | Open-Domain Dialogue |
| dialogpt_large-typical_02-dev | Cosine distance | 0.073281 | Open-Domain Dialogue |
| dialogpt_large-typical_095-dev | Cosine distance | 0.074061 | Open-Domain Dialogue |