# OpenReview forum: "What Comes Next? Evaluating Uncertainty in Neural Text Generators Against Human Production Variability"
_EMNLP/2023/Conference — EMNLP 2023 Main_

### Official Review · Reviewer_mAJR · 2023-07-31

**Soundness:** 4

**Excitement:**

4: Strong: This paper deepens the understanding of some phenomenon or lowers the barriers to an existing research direction.

**Paper Topic And Main Contributions:**

Summary: The authors propose a method for quantifying the uncertainty of natural language generation systems and compare a number of current NLG systems against humans. They find that for open-ended tasks, models demonstrate more uncertainty than people, while this is reversed for more closed tasks, such as translation. They argue that decoders make models’ generation more human-like but that no decoder is stand-out compared to the others. Overall I thought this paper is on a very interesting subject, however the paper could be improved through more careful justification of their probe choices.

**Questions For The Authors:**

Questions/Comments (line number)

A (55) Should an NLG system capture the observed variability? Why? What’s wrong with an NLG system that gives, say, 5 excellent continuations to a prompt when humans can reliably come up with 10?

B (112) What is “statistical fitness” in this context? Is this a technical term?

C (186) What is a probability distribution over a production? Typically, LMs give probability distributions over strings. Is that what you mean here?

D (197) For some previous work on how to estimate things like entropy from linguistic distributions, see Aurora et al., (2022) “Estimating the Entropy of Linguistic Distributions”

E (209) My immediate first thought here was that you were going to look at length as one of the probe metrics. Why didn’t you look at length? It seems like a relevant factor along which production can vary.

F (245) This is a relatively simple metric for looking at syntactic complexity. Did you consider others that take syntactic structure into account, such as, say Tree Edit Distance?

G (436) I think it’s necessary to introduce the Wasserstein 1 Distance formally in the main body of the text, given that it’s not a well known measure of distance.

**Reasons To Accept:**

Reasons to Accept:

(a)  The topic of this paper is both interesting, understudied and (to me at least) timely.

(b) The paper has two clear take-aways which will be relevant to the community: (a) underestimation of variability for closed tasks and overestimation for open tasks and (b) the effect of decoding strategies on the variability.


**Reasons To Reject:**

Reasons to Reject:

(a) The probing process could be justified better, and explained more clearly. In fact, I think a lot of good justification already exists in Appendix A. I would suggest moving this discussion into the main body of the text and moving Section 7 into an Appendix.

(b) Overall, I think the authors could draw more connections to the psycholinguistic literature, especially to speech planning and production.

(c) Section 6.2 is methodologically weaker than the other experiments in my opinion. The authors conclude that the decoding settings are close to the unbiased sampling which are all in the “same ballpark” as human scores. The authors need to figure out how to quantify this claim rigorously.



**Reproducibility:**

2: Would be hard pressed to reproduce the results. The contribution depends on data that are simply not available outside the author's institution or consortium; not enough details are provided.

**Reviewer Confidence:**

4: Quite sure. I tried to check the important points carefully. It's unlikely, though conceivable, that I missed something that should affect my ratings.

**Typos Grammar Style And Presentation Improvements:**

(102) I would mention locally typical sampling here. The whole point is to mimic human production (variability) so informed readers will be wondering about it at this point.

(117) To me, it’s still unclear exactly what you did, at the end of the introduction. I would state these in more mechanistic terms during the intro, especially the probe itself.

(308) This looks like entropy, which really threw me off for a second. I guess you use H for human here? I would suggest changing it.
(359) “an text generator” → “a text generator”

(Footnote 8 / Figure 5) I would mention the text of Footnote 8 in Figure 5. I was very confused about why there was no human data presented in this plot at first.

(593) Overall, I think the discussion of aleatoric uncertainty can be dropped. This was mentioned briefly in the beginning but not given a formal treatment and I don’t think it adds much to the understanding of the results.

---

> ### Author Rebuttal · Authors · 2023-08-23
>
> Thank you for so thoroughly engaging with our work!
>
> We address your concerns next.
>
> (a) Thank you for your suggestion, we will add a condensed version of Appendix A to the main paper!
>
> (b) While we are very excited about the deeper connections between production variability and its sources, the format of a conference paper would not have allowed us to do them full justice. We have therefore decided to keep our contribution in this paper focused on the proposed statistical framework and on showcasing its potential for the evaluation and analysis of NLG systems (hence the submission to the “Interpretability, Interactivity, and Analysis of Models for NLP” track). We explore this connection further in two follow-up studies, which we look forward to referring to in the camera-ready of the paper.
>
> (c) Thanks for making this point, which we believe highlights an issue with the presentation of our quantitative results (we will remove phrases like “same ballpark” from the camera-ready) rather than with their substance. Indeed, the very reason for the introduction of human control groups (lines 439-466) was to be able to rigorously interpret (absolute) divergence scores relative to control divergence scores obtained by human held-out groups. We will consider expressing the relation between models’ and control groups’ divergence scores mathematically (e.g., through a ratio), though we are not sure this will make results more interpretable or informative.
>
> We address your questions next, and will make sure the camera-ready reflects our answers.
>
> (A) In this paper, we evaluate NLG systems in terms of whether they are faithful models of human production, which is something that allows us to apply them to interesting psycholinguistic questions like the one hinted to in question-response (b). From a more applied NLP perspective, there is nothing inherently wrong with a system that shows higher or lower levels of variability than humans. In fact, for some applications (e.g., fact-based question answering), lower levels of variability than those observed in a population of humans may be desirable. Regardless of application-specific target levels of variability, our framework provides the technical tools to measure variability and compare it to those targets.
>
> (B) We used "fitness" to refer to a statistical model’s ability to *fit* or *reproduce* (characteristics of) a set of observations. We will use "model fit" instead.
>
> (C) Yes, for simplicity, we equated linguistic productions to their string representation.
>
> (D) We will include a mention to this interesting work in our camera-ready.
>
> (E) We have indeed used length difference as a similarity metric in preliminary experiments, but we decided to focus on our three interpretable metrics that target three classic levels of linguistic analysis. We hope that future work will investigate length differences as well as other dimensions of interest (see lines 678-685 in the Limitations section for further examples).
>
> (F) The space of possible similarity metrics is very large and exploring it thoroughly was beyond the scope of this paper. Syntactic distances can indeed be computed using metrics that capture structural differences between utterances in a more fine-grained manner (e.g., tree edit distance or difference in syntactic tree depth). Similarly, semantic distances can be computed with a more taxonomical approach (e.g., via WordNet synsets) or using NLI models to capture semantic equivalence (Kuhn et al. 2023, as cited in the paper); and distances between dialogue act types can be detected using dialogue act classifiers (Stasaski and Hearst, as cited in the paper).
>
> G) If space allows, we will expand and move our discussion of W1 in Appendix B to the main text.
>
>
> **Reproducibility**. We accidentally removed mention of code in our submission, but we will most definitely release the code required to replicate all our experiments. To reiterate our response to R1; in the camera-ready, we will include a link to a public repository with the code to reproduce all experiments in our paper. To further aid reproducibility, (1) we will upload our fine-tuned models to the Hugging Face model hub and (2) we are considering the additional release of the large dataset of model generations used in our study (if this release does not infringe the licences of the four NLG corpora under analysis, we will make the data available through an open data library like Zenodo). We hope this will change your thoughts on the reproducibility of our work.

---

### Official Review · Reviewer_hk61 · 2023-08-03

**Soundness:** 3

**Excitement:**

4: Strong: This paper deepens the understanding of some phenomenon or lowers the barriers to an existing research direction.

**Paper Topic And Main Contributions:**

This work empirically assesses the production (output text) variability for between text generation model and human productions. The product variability is defined by different similarity functions in lexical, syntactic, and semantic levels. The paper considers the human variability as a form of aleatoric uncertainty, and compares model variability with human variability in divergence of such variability metrics.
The experiments are conducted on four NLG tasks with each having multiple references, and the results show that NLG models overestimate production variability in open-ended tasks and underestimate it in more constrained tasks. Popular decoding algorithms for generation model are demonstrated to have a similar effect on the difference between model and human production variability. Additionally, qualitative analysis on instance-level production variability is also complemented to gain insights on model variability.

**Questions For The Authors:**

- Q1: What's the relationship between the cross-variability in eq3 and divergence metrics used in section 6.1 and 6.2? Why is the marginalized cross-variability not used directly as statistical divergence?
- Q2: I could not fully grasp the part of "variability of lack of situational grounding" from line 561. Why is the context example in line 571-272 not available to model in prompt, but is available to human (I assume so)? Does it mean that the input information, i.e. prompts to both human and model, is different? Is the main point on model-human comparison or human-human comparison as mentioned in the beginning?

**Reasons To Accept:**

- S1: The problem of uncertainty comparison between model and human is less explored, and I think the paper could spark more interest in this domain.
- S2: The definition of production variability on the generated texts with statistical functions could serve as a new measurement to the NLG uncertainty/diversity.
- S3: The paper is well-written and very easy to follow.

**Reasons To Reject:**

- W1: The biggest problem to me is the the selection of NLG models for different tasks. For each task, a different model is picked without justifications of the reason and rationale of such selection. Furthermore, the same model is not evaluated across tasks, and the single model in each task is not compared with other alternatives at all. This could cause concerns on the reliability of results and conclusion.
- W2: The divergence metrics are different in section 6.1 and 6.2. Although the authors provide explanations of the two, the switch of the metrics might imply that different metrics could have a considerable impact on the conclusion.

**Reproducibility:**

3: Could reproduce the results with some difficulty. The settings of parameters are underspecified or subjectively determined; the training/evaluation data are not widely available.

**Reviewer Confidence:**

4: Quite sure. I tried to check the important points carefully. It's unlikely, though conceivable, that I missed something that should affect my ratings.

**Typos Grammar Style And Presentation Improvements:**

- P1: I think it is very confusing for the definition of k(.,.) It is called production variability, but essentially a similarity metric. Therefore, higher value of k(., ,) indicates low variability, and vice versa. In multiple places, e.g. line 237-238 and Figure 2, the name and the values are misleading, and I suggest the authors could change it to a more consistent definition like (1 - similarity).

---

> ### Author Rebuttal · Authors · 2023-08-23
>
> Thank you for your comments and questions.
>
> **W1 “selection of models”**
> For each of the four NLG tasks in our experiments, we have selected models that (1) are publicly available, (2) have been used in previous work _on the task_, (3) are conventionally accepted as good models _for the task_ and (4) are reasonably sized. For text simplification, we did not find any available trained model, so we used a versatile model like FlanT5. We agree it is important to include this explanation of the rationale used to select models and we will do so in the camera-ready.
>
> Please also allow us to clarify that, with this paper, we are proposing a framework for probing the uncertainty of NLG models’ and evaluating it against human productions, and this contribution is intended to go beyond the experimental results obtained on our selection of models. We hope that our framework will be used by the community as an additional important evaluation criterion for NLG systems, especially to assess them in more open-ended tasks. In the camera-ready, we will also clarify the relative importance of the framework vs. the empirical results.
>
> **W2 “Different divergence metrics in 6.1 and 6.2 may impact conclusions”**
> We report results for all divergence metrics in Appendices D and F. These confirm that the trends we highlight in the main paper hold regardless of the specific divergence metric used. We mention this explicitly in Section 6.1 (lines 419-422) and more succinctly in Section 6.2 (lines 436-438). As you noted, we explain why we introduce each divergence metric (lines 398-407 and 433-436). We will use the additional page of the camera-ready to briefly discuss the nuanced differences in results and their interpretation obtained with mean difference vs. Wassterstein distance, which (as you can judge from the Appendix) do _not_ have a significant impact on our conclusions.
>
> **Q1 “Cross-variability in eq3 vs divergence metrics in section 6.1 & 6.2; why not use marginalized cross-variability directly as statistical divergence?”**
> Thank you for this thoughtful question. We tried to address this point in lines 374-386 but we will have another go here, hoping this clarifies our rationale.
>
> It is certainly reasonable to evaluate a model based on what you call “marginalized cross-variability”. This is the interpretation exploited by most automatic NLG metrics: large positive cross-variability values indicate that model generations closely match the human references.
>
> However, in this paper, we do not take any single production as a ‘reference’ to be closely ‘matched’. This is because references can vary a lot with respect to each other (especially in open-ended task), as we try to point out throughout the paper and as we show empirically in Section 5. It is exactly this variability that we target in our analyses, and in particular, models’ ability to reproduce human-like levels thereof.
>
> So, when we compute cross-variability following Eq. 3, we use the human responses not as ‘references’ to be ‘matched’ but as a set of plausible responses with which we estimate model variability. This complements our estimates of (self-)variability in Eq. 2.
>
> To evaluate whether estimates of model variability (both obtained with Eq. 2 and Eq. 3) are in compliance with the variability observed in humans (Eq. 1), we use two metrics of statistical divergence.
>
> **Q2 "lack of situational grounding?”**
> The WritingPrompts dataset is constructed from a Reddit forum where one user posts the beginning of a story (or a story prompt), and other users post continuations of that story (or responses to that story prompt). For example:
>
> Prompt: “You're a king and your kingdom is in the midst of rebellion. The issue is that you don't disagree with the rebellion.”
>
> Response 1: “I’m the king. I should be kicking and crying and cursing the lands for this rebellion… but… I’m not mad. These are my people, and they have every right to be mad by my standing [...]”
>
> Response 2 (*one hour later*): “I am the supreme sovereign. The highest power. The heart and soul of the realm. None dare question me, and all do as I say. Or at least that's the story we present to the. The reality is much less clear cut. In the real world [...]”
>
> Response 3: …
>
> etc.
>
> Just like the humans in this example, the NLG system in our experiments is conditioned on the prompt—and on the prompt only.
>
> Now to the example of "variability of lack of situational grounding" from line 561. The prompt for this instance is “all top level comments in this prompt take place in the same world, so make them all fit together”. That is, the user who proposes this prompt is asking other users to make use of each others’ responses (in particular, “top level comments”) to create stories that feature elements taken from multiple user responses. As a consequence, while the first few responses are likely only conditioned on the prompt (so quite open-ended), later responses will also contain elements from earlier user comments, which users can see (and are thus less open-ended). The NLG system, instead, can only see the prompt and therefore lacks the additional situational grounding that Reddit users can take into account.
>
> *"Is the main point on model-human comparison or human-human comparison as mentioned in the beginning?"* We evaluate model outputs against human responses. The human-human similarities are necessary to obtain an estimate of plausible variability to evaluate the models against.
>
> **P1 “variability or similarity”**
> We had many internal discussions about this point. We hoped variability would allow readers to more easily connect to and reason about uncertainty (high variability = high uncertainty). We do agree that the current presentation can be improved further, and will consider either changing the underlying similarity functions to distance functions, or renaming our production variability probes k(.) to production similarity probes.

---

### Official Review · Reviewer_haSd · 2023-08-04

**Soundness:** 4

**Excitement:**

3: Ambivalent: It has merits (e.g., it reports state-of-the-art results, the idea is nice), but there are key weaknesses (e.g., it describes incremental work), and it can significantly benefit from another round of revision. However, I won't object to accepting it if my co-reviewers champion it.

**Paper Topic And Main Contributions:**

The paper tackles the problem of uncertainty in NLG models against human production variability. More specifically, uncertainty representation is the probability distribution over sequences of tokens from two mechanisms - an underlying statistical model and an interactive decoding algorithm. The uncertainty representation is analyzed in compliance with the variation of human production and categorized as aleatoric or irreducible uncertainty within the data generation process.

This study analyzes principal applications within NLG, such as text classification, machine translation, text simplification, and conversational AI.

Based on the experimental results of the statistical fitness (calibration of models’ uncertainty), the authors conclude that NLG models tend to overestimate the production variability in open-ended tasks and the opposite in open-ended ones.


**Reasons To Accept:**

This is important and interesting research about NLG evaluation and addressing this problem adequately would lead to a beneficial impact within NLG criteria for evaluating language models.

This provides a valid tool for the NLG evaluation, as well as, adds value to NLG models' calibration.


**Reasons To Reject:**

The uncertainty and variability terms seem to mislead the benefits of this study, and also reduce the paper’s readability. Another term that reduces the paper’s impact is “fitness” instead of “model fit”.

Experimental results seem to benefit open-ended information as reported by the authors.

The authors did not mention the reproducibility of the work and if the code will be available for further research.
Ethical considerations are not highlighted within the paper.


**Reproducibility:**

3: Could reproduce the results with some difficulty. The settings of parameters are underspecified or subjectively determined; the training/evaluation data are not widely available.

**Reviewer Confidence:**

3: Pretty sure, but there's a chance I missed something. Although I have a good feel for this area in general, I did not carefully check the paper's details, e.g., the math, experimental design, or novelty.

**Typos Grammar Style And Presentation Improvements:**

*Typos:*

131 i.e. instead of i.a.

148 add connector: and length)

170 add space between words: strategies—yet

171 add space between words: )—or

404 missing comma for i.e., - overestimate variability, i.e.

454 add space between punctuation: ).Upon

Figures 12 and 13 have the same information. Also, please add to the caption what is presented in each figure.

649 responses.

658 remove purposes.

659 remove purposes.

664 humanly-acceptable?

672 e.g..,

675 correct reference to "see Levinson (1983)".


*Readability suggestions:*

144 (e.g., Yuan et al., 2021; Zhong et al., 2022).

132-134. Instead of just referencing BARTscore, better adding BartScore (Yuan et al., 2021), so the reader would understand better, as it was done on lines

163 change "traction".

163 remove e.g.

196 change given an input: about the output Y was given input X = x.

203 change usually enough.

210 remove the apostrophe from process'

210 to 212 - please rephrase due to inputs, outputs.

212 to 216 = rephrase.

Footnote 2: rephrase - how it is hard to estimate and interpret for text generators.

Figure 1: Remove the apostrophe of humans.

Figure 1: add boldfaced examples in the text. Figure 1 is difficult to read due to the flow of the text. It is not clear which one is generated
by DialoGPT.

424 to 425: please rephrase.

Footnote 6: Add italics to pmf and refer to the appendix section. Add "that" to which is all that our analysis tools require.

430 to 431: "handful of cases" to "a sample".

433. Change "instead", or possibly rephrase.

448 to 454: change appreciate, rephrase paragraph.

488 Change "now" or rephrase.

Figure 9: Tuples that share "the same" color have different decoding parameters.

511-521: Change the text for a table or a different form to improve readability.

552-553: rephrase

571-572, remove parentheses and change the underline for italics.

587 change "only a single generation" for "a single generation".

1131 because (a)

1133 and (b)

Figures 6 and 7 have the same caption.

Figure 8 (higher means pairwise similarity).

Figure 9 for tasks, probe, and decoding algorithm tuples.

Figure 9 Tuples that share "the same" color.

Figures 16 and 17 have the same caption.

---

> ### Author Rebuttal · Authors · 2023-08-23
>
> Thank you for such a detailed review! We are happy that you consider our framework as a valuable tool for NLG evaluation, and we are confident we can address your comments in the camera-ready version of the paper.
>
> We will incorporate your editing suggestions and improve our terminology. In particular, we will use the additional page to clarify the connection between uncertainty and variability. As we attempt to explain in Section 1 (lines 68-76), the term variability implies that there are multiple possible responses given an input. An NLG model's representation of uncertainty should ideally reflect this with a wide distribution over the strings that are plausible according to a population of speakers. Thus, the term “variability”  is closely related to aleatoric or data uncertainty, which is uncertainty inherent to the data that a model is trying to capture, rather than the model itself (this is often called epistemic uncertainty).
>
> We used “fitness” to refer to a statistical model’s ability to fit or reproduce (characteristics of) the data. We will use “model fit” instead.
>
> We are also happy to include a discussion of your comment _“Experimental results seem to benefit open-ended information as reported by the authors.”_ if you please clarify for us why you perceive this as a concern rather than as an empirical finding.
>
> Reproducibility is important to us. In the camera-ready, we will include a link to a public repository with the code to reproduce all experiments in our paper. To further aid reproducibility, (1) we will upload our fine-tuned models to the Hugging Face model hub and (2) we are considering the additional release of the large dataset of model generations used in our study (if this release does not infringe the licences of the four NLG corpora under analysis, we will make the data available through an open data library like Zenodo).
>
> Finally, we did not think the paper required an ethical consideration section, but we have provided a detailed limitation section. We are willing to include any suggestions you might have for an ethical consideration section in the camera-ready version of the paper.

---

### Meta-Review · Area_Chair_y2T3 · 2023-09-19

**Recommendation:** 5

**Metareview:**

The authors analyze how NLG models represent uncertainty. The authors argue that NLG systems should capture variation within human language production.For every generation prompt text, the authors compare properties of generations obtained through repeated model sampling, with multiple human productions— finding that uncertainty is lower when the task setting is more constrained. All authors appreciated the motivation of this work and recognize the need for improved calibration methods.

---

### Decision · Program_Chairs · 2023-10-07

**Decision:**

Accept-Main

**Comment:**

The authors analyze how NLG models represent uncertainty. The authors argue that NLG systems should capture variation within human language production.For every generation prompt text, the authors compare properties of generations obtained through repeated model sampling, with multiple human productions— finding that uncertainty is lower when the task setting is more constrained. All authors appreciated the motivation of this work and recognize the need for improved calibration methods.